

# Investigating the quality of modeled aerosol profiles based on combined lidar and sunphotometer data

Nikolaos Siomos[1], Dimitris S. Balis[1], Anastasia Poupkou[1], Natalia Liora[1], Spyridon Dimopoulos[1], Dimitris Melas[1], Eleni Giannakaki[2,5], Maria Filioglou[1,2], Sara Basart[3], and Anatoli Chaikovsky[4]

[1]Laboratory of atmospheric physics, Physics Department, Aristotle University of Thessaloniki, Greece
[2]Finnish Meteorological Institute, Atmospheric Research Centre of Eastern Finland, Kuopio, Finland
[3]Earth Sciences Department, Barcelona Supercomputing Center, BSC-CNS, Barcelona, Spain
[4]Institute of Physics, National Academy of Science, Minsk, Belarus
[5]Department of Environmental Physics and Meteorology, Faculty of Physics, University of Athens, Greece

*Correspondence to:* N. Siomos (nsiomos@physics.auth.gr)

**Abstract.** In this study we present an evaluation of the Comprehensive Air Quality Model with extensions CAMx for Thessaloniki using radiometric and lidar data. The aerosol mass concentration profiles of CAMx are compared against the fine and coarse mode aerosol concentration profiles retrieved by the Lidar-Radiometer Inversion Code LIRIC. The CAMx model and the LIRIC algorithm results were compared in terms of mean mass concentration profiles, center of mass and integrated mass

concentration in the boundary layer and the free troposphere. The mean mass concentration comparison resulted in profiles within the same order of magnitude and similar vertical structure for the fine particles. The mean center of mass values are also close with a fractional bias of 24.8%. On the opposite side, the coarse mode appears to be underestimated by the model below 4km and overestimated above. In order to grasp the reasons behind the discrepancies, we investigate the effect of aerosol components that are not properly included in the model's emission inventory and boundary conditions such as the wildfires

and the desert dust component. The identification of the cases that are affected by wildfires is performed using wind backward trajectories from the Hybrid Single Particle Lagrangian Integrated Trajectory Model HYSPLIT in conjunction with satellite fire pixel data from the MODerate-resolution Imaging Spectroradiometer (MODIS) Terra and Aqua global monthly fire location product MCD14ML. By removing those cases the correlation coefficient improves from 0.44 to 0.86 for the fine mode integrated mass in the boundary layer. The fine mode center of mass fractional bias also decreases to 16.9%. Concerning the

analysis on the desert dust component, the simulations from the updated version of the former Dust Regional Atmospheric Model called BSC-DREAM8b were deployed. When only the desert dust cases are taken into account, BSC-DREAM8b generally outperforms CAMx when compared with LIRIC, achieving a correlation of 0.91 and a fractional bias of -18.9% for the integrated mass in the free troposphere and a correlation of 0.44 for the center of mass. CAMx, on the other hand, both underestimates and anti-correlates the integrated mass in the free troposphere. Consequently, the accuracy of CAMx is limited

concerning the transported dust cases. We conclude that the performance of CAMx appears to be best for the fine particles, especially in the boundary layer. At the same time it systematically fails to successfully predict the coarse mode. Sources of particles not properly taken into account by the model are confirmed to negatively affect its performance.





# 1 Introduction

There is a wide variety of atmospheric models that are capable of providing vertical profiles of the aerosol mass concentration (e.g. CAMx, BSC-DREAM8b, European Monitoring and Evaluation Programme EMEP, LOng Term Ozone Simulation - EURopean Operational Smog model LOTOS-EUROS, CHIMERE). This is achieved through simulation of the atmospheric

motion and the chemical reactions that are taking place inside the atmosphere. The most common approach to validate the modeled vertical mass concentration products is to compare with surface and columnar concentration or optical measurements either from ground-based or satellite instruments (e.g., Takemura et al., 2002; Stier et al., 2005; Katragkou et al., 2010; Huneeus et al., 2011; Basart et al., 2012a; Marécal et al., 2015). This approach, however, doesn't verify the ability of the model to accurately predict the vertical distribution of the aerosol concentration. Observational aerosol profiles comparable with the

modeled ones are required for this purpose. Remote sensing techniques such as lidar measurements can provide us with this sort of profiles. Since the main lidar products typically involve optical aerosol properties such as the aerosol backscatter and extinction coefficient profiles, it is common to ensure comparability by converting the model's output after applying appropriate techniques. For example Mona et al. (2014) compare between the dust extinction profiles of the BSC-DREAM8b model and the respective EARLINET (European Aerosol Research LIdar NETwork) profiles for a 12 year period in Potenza. Meier et al.

(2012) use lidar backscatter profiles as one of the tools to evaluate the Consortium for Small-scale Modeling - Multi-Scale Chemistry Aerosol Transport (COSMO-MUSCAT) model for the PM2.5 and PM10 particles. Hodzic et al. (2004) recreate the lidar attenuated backscatter profiles using the output of the CHIMERE model in order to compare the model's PM10 profiles with the lidar measurements.

On the other hand, there are techniques that allow the estimation of the aerosol vertical concentration from remote sensing

lidar measurements using a suitable algorithmic inversion method (e.g., Böckmann, 2001; Veselovskii et al., 2002; Raut and Chazette, 2009; Lopatin et al., 2013; Chaikovsky et al., 2016). The advantage of this approach is that the modeled product can be directly validated without the need of conversion. The literature focused on the validation of dust transportation models with observational aerosol concentration profiles is quite mature. For example, Binietoglou et al. (2015) have presented a methodology based on LIRIC to evaluate the performance of dust models using data from multiple AERONET (AErosol

RObotic NETwork) and EARLINET stations. Granados-Muñoz et al. (2016) also use the LIRIC algorithm to compare between observational data and a variety of dust models in the frame of July 2012 CHemistry and AeRosols Mediterranean EXperiments ChArMEx/EMEP campaign. However, there is a lack of studies that focus on the evaluation of both PM2.5 and PM10 concentration profiles simulated by atmospheric models. Royer et al. (2011) compare the simulations of the CHIMERE chemistry transport model for the PM10 particles using lidar PM10 concentration profiles derived with the methodology of

Raut and Chazette (2009). In this study we investigate the validity of the aerosol concentration profiles simulated with the air quality model Comprehensive Air Quality Model with extensions (CAMxversion5.3), (ENVIRON 2010, 2010) for Thessaloniki, Greece (40.5N, 22.9E) using the results of the Lidar/Radiometer Inversion Code (LIRIC). Instead of evaluating the performance of CAMx only for the PM10 particles, we separate the fine from the coarse aerosols and perform the validation for each mode individually.





CAMx is running operationally to produce a 3-day air quality forecast for Thessaloniki (Zyryanov et al., 2012; Marécal et al., 2015). It provides vertical concentration profiles of a variety of gaseous and aerosol components.

A second model, the desert dust transportation model BSC-DREAM8b (Nickovic et al., 2001; Pérez et al., 2006a, b; Basart et al., 2012b) has also been included in the analysis in order to investigate the performance of CAMx in the case of dust transportation events. This model can provide total dust concentration profiles.

The LIRIC inversion (Chaikovsky et al., 2016), on the other hand, is a technique used to estimate the concentration profiles of the fine and coarse mode aerosol using both sunphotometer and lidar data. Lidar and sunphotometer measurements performed at the Laboratory of Atmospheric Physics of the Aristotle University of Thessaloniki, Greece (40.5N, 22.9E) from the period 2013-2015 were used as input data for the algorithm.

Validating the accuracy of CAMx simulations for Thessaloniki for the period 2013-2015 could prove useful in the aerosol classification procedure of the lidar measurements since individual aerosol components are provided by CAMx. Furthermore, from the modelers' point of view, the comparison could also reveal the need of adjustments in the model's aerosol emissions, boundary conditions and mixing processes.

The paper is organized as follows. In section 2 the two models, CAMx and BSC-DREAM8b, and the LIRIC algorithm are described in detail. The 3rd section is devoted to the methodology of the analysis. This includes the preprocessing of the lidar and the sunphotometer measurements and the characterization of the lidar profiles, the demonstration of the strategy that we applied for the comparison and the application of two example cases. The results of the study are discussed in section 4. Finally, section 5 contains the main conclusions of this study.

## 2  Data, Algorithm and Models

### 2.1  The lidar system of Thessaloniki

Lidar measurements from the THEssaloniki LIdar SYSstem (THELISYS), that is located at the Laboratory of Atmospheric Physics (LAP) of the Aristotle University of Thessaloniki ($40.5^o$ N, $22.9^o$ E) at 50m above sea level, during the period 2013-2015 were selected for this study. The setup of the system in this period includes two raman channels at 355nm and 532nm and three elastic channels at 355, 532 and 1064nm. The raw lidar signals from the elastic channels at the three aforementioned wavelengths are necessary in order to perform the LIRIC inversion. All signal pre-prepossessing procedures are applied directly in the LIRIC algorithm. The lidar station of Thessaloniki participates in the European Aerosol Research Lidar Network (EARLINET) Schneider et al. (2000) since 2000. More details on the instrument can be found in Amiridis et al. (2005) and (Giannakaki et al., 2010).

### 2.2  The CIMEL sunphotometer

In order to apply the LIRIC inversion, sunphotometer observations are necessary. The CIMEL multiband sun-sky photometer of Thesaloniki was deployed for this purpose. The instrument participates in the AERONET Global Network. It belongs to the



Laboratory of Atmospheric Physics (LAP) and it is located in a distance less than 50m from the lidar instrument (see section 2.1) in the same altitude. It automatically performs direct solar irradiance and sky radiance measurements at 340, 380, 440, 500, 670, 870, and 1020nm. The data processing is performed automatically with the AERONET inversion algorithms (Dubovik and King, 2000; Dubovik et al., 2006). The level 2 Version 2 Inversion data during the period 2013-2015 were used in this

study. The instrument and the AERONET infrastructure are described in detail in Holben et al. (1998).

## 2.3  The Comprehensive Air quality Model with extensions CAMx

An air quality forecast modeling system was set-up in the framework of the EU Monitoring Atmospheric Composition and Climate (MACC) project. It consists of the meteorological model Weather Research and Forecasting model (WRF version 3.5.1) described in Skamarock et al. (2008) and the photochemical model CAMx (version 5.3). It is designed to provide the air

quality forecast in four nested grids covering Europe (30 km spatial resolution European grid covering also a part of Sahara desert), the Eastern Mediterranean (10 km spatial resolution grid) and the Greek urban centers Thessaloniki and Athens (2 km spatial resolution grids). A nesting technique is applied in order to increase the accuracy at the area of interest i.e. Thessaloniki. A single grid point in Thessaloniki is chosen at (40.633 N, 22.956 E) for the model outputs processed in the present study to coincide with the lidar measurements. The model grids are configured in 17 vertical layers extending up to about 9.5 km above

ground level (agl). The temporal resolution of CAMx outputs and consequently of the simulated profiles is one hour.

The aerosol particles are modeled using a static two-mode coarse/fine scheme in CAMx for the representation of the particle size distribution. Fine particles have an aerodynamic diameter smaller than 2.5 μm while coarse particles have an aerodynamic diameter higher than 2.5 μm and smaller than 10 μm. A total of 20 individual aerosol components plus the aerosol water content that is absorbed by the hygroscopic particles are provided by the model (see table 1). The aerosol aqueous inorganic chemistry

is applied according to the RADM-AQ aqueous chemistry algorithm (Chang et al., 1987). The partitioning of the inorganic aerosol constituents between the gas and aerosol phases is performed using the ISORROPIA thermodynamic module (Nenes et al., 1998). The secondary organic aerosol (SOA) formation/partitioning was performed with the use of the SOAP scheme (Strader et al., 1999). SOA are formed by non-methane volatile organic compounds of anthropogenic and natural origin. Details on the CAMx aerosol components such as the hygroscopicity and the fine or coarse mode type can be found in table 1.

CAMx is applied with the use of gaseous and particulate anthropogenic and natural emissions. Particulate matter emissions from natural sources (windblown dust and sea salt aerosol) and biogenic volatile organic compounds from vegetation are estimated using the Natural Emission MOdel NEMO version 1 (Markakis et al., 2009; Poupkou et al., 2010; Markakis, 2010) driven by the WRF meteorology. The Model for the Spatial and Temporal Distribution of Emissions (MOSESS) (Markakis et al., 2013) was applied for the calculation of spatially and temporally disaggregated and chemically speciated anthropogenic

emission data of the following pollutants: CO, NOx, SO2, NH3, NMVOC, PM10 and PM2.5. The anthropogenic emissions were estimated using either activity data with methodologies and emission factors of the EMEP/CORINAIR - CORe INventory AIR emission inventory guidebook (EEA, 2006) or the emission database of The Netherlands Organization (TNO) for the reference year 2007 (Kuenen et al., 2011). Anthropogenic emissions for the following sources are accounted for: energy production, central heating, industry, transportation, waste treatment and disposal, agricultural activities (i.e. biomass burning,





fertilization), extraction and distribution of fossil fuels. It is important to mention that particle emissions due to dust resuspension from agricultural activities and road traffic as well as the wildfire emissions are not currently included in CAMx simulations. Saharan dust emissions are taken into account only indirectly in the CAMx chemical boundary conditions provided by the global forecast modeling systems Integrated Forecasting System IFS-MOZART until October 2014 and C-IFS

afterwards (Flemming et al., 2009; Morcrette et al., 2009; Stein et al., 2012) in the framework of the MACC project. The model has been evaluated during the MACC-II project (Marécal et al., 2015) and also with observations from the National Monitoring Network of the Air Quality (Melas et al., 2017).

## 2.4   The BSC-DREAM8b model

The transported desert dust particles and the forest fire particles are common categories of aerosol components in Thessaloniki

(Amiridis et al., 2005; Basart et al., 2009; Giannakaki et al., 2010). As it has already been stated, the setup of CAMx in Thessaloniki includes the desert dust component only from the global boundary conditions and it doesn't include wildfire emissions at all. Biases in those two aerosol components are expected to affect both aerosol modes since the desert dust particles are coarse dominant (Shettle and Fenn, 1979; d'Almeida, 1987) and the biomass burning particles are fine dominant (Tesche et al., 2009; Groß et al., 2013).

The desert dust component can be further analyzed by comparing LIRIC with a dust specialized model. The dust transportation model BSC-DREAM8b (Pérez et al., 2006a, b; Nickovic et al., 2012; Basart et al., 2012b) was chosen for the comparison. BSC-DREAM8b is managed by the Barcelona Supercomputer Center (BSC) and provides operational forecasts since May 2009, and is also participating in the northern Africa–MiddleEast–Europe (NA-ME-E) node of the World Meteorological Organization (WMO) Sand and Dust Storm Advisory and Assessment System (SDS-WAS) project. The BSC-DREAM8b model

is embedded into the Eta/NCEP atmospheric model and solves the mass balance equation for dust, taking into account the different processes of the dust cycle (i.e., dust emission, transport and deposition). The updated version of the model includes 8 particle size bins (0.1–10 µm radius range) and dust-radiative feedback.

  The BSC-DREAM8b model has been evaluated for longer periods over northern Africa and Europe (e.g., Jiménez-Guerrero et al., 2008; Pay et al., 2010; Pay et al., 2012; Basart et al., 2012b, a; Gama et al., 2015) and against experimental campaigns

in source regions during the SAharan Mineral dUst experiMent SAMUM-1 (Haustein et al., 2009) and the Bodélé Dust Experiments (BoDEx, Todd et al., 2008). Furthermore, daily evaluation of BSC-DREAM8b with near-real-time observations is conducted at BSC. Currently, the daily operational model evaluation includes satellites (MODIS and MSG - Meteosat Second Generation) and AERONET sun photometers. Some comparisons between lidar and forecast models profiles were performed in terms of aerosol vertical distribution for specific Saharan dust events in the Mediterranean Basin (e.g., Balis et al., 2004;

Pérez et al., 2006a; Amiridis et al., 2009; Mona et al., 2012; Gobbi et al., 2013; Amiridis et al., 2013; Mona et al., 2014). In addition, Binietoglou et al. (2015) includes BSC-DREAM8b as one of the models that participate in their analysis validating its performance against LIRIC retrievals in 10 EARLINET/AERONET stations.

  The present analysis includes the daily runs of BSC-DREAM8b. The initial state of dust concentration in the model was defined by the 24 h forecast from the previous-day model run. The Final Analyses of the National Centers of Environment



Prediction (NCEP/FNL; at $1^o \times 1^o$) at 0UTC were used every 24 hours as initial conditions and boundary conditions at intervals of 6h. The model configuration used for the present study includes 24 Eta vertical layers extending up to approximately 15 km in the vertical. The resolution is set to $0.3^o$ in the horizontal. The temporal resolution of the simulations is 3 h. The domain of simulation covers northern Africa, the Middle East and Europe. It is worth mentioning that re-suspended wind-blown dust and

the considered desert dust sources are limited to northern Africa and the Middle East ($< 35^o$ N) in the BSC-DREAM8b model.

## 2.5 The LIdar-Radiometer Inversion Code LIRIC

The LIRIC algorithm utilizes both radiometric data that have been processed by the AERONET inversion algorithm (Dubovik and King, 2000; Dubovik et al., 2006) and also raw lidar signals at three wavelengths (355 nm, 532 nm and 1064 nm) in order to estimate the aerosol concentration profiles for the fine and coarse particles. The radiometric data used as input includes

the aerosol size distribution, the aerosol volume concentration in the two modes (fine and coarse), the aerosol optical depth (AOD) and the single scattering albedo (SSA) of each mode, the complex refractive index in each wavelength, the sphericity and the aerosol phase function. The minimum of the aerosol size distribution in the 0.194 - 0.576 μm radius range is used as the size boundary that separates the fine from the coarse mode. In case that the particle depolarization ratio is also provided, it is possible for the algorithm to further separate the spherical coarse particles from the non spherical ones. Nevertheless, only

the fine and coarse mode retrievals are taking part in this study due to the lack of depolarization ratio profiles.

The algorithm main products are the vertical volume concentration profiles in the two modes. In brief, the algorithm searches for the profile per mode that gives the best agreement between the actual data and the reconstructed data from the algorithm, also demanding a certain degree of vertical smoothness in the final product. The reconstructed data include the aerosol backscatter profiles in the three lidar wavelengths and the columnar volume concentration values per mode. A detailed description of LIRIC

can be found in Chaikovsky et al. (2016).

The effects of multiple user defined uncertainties on the final result has been studied by Granados-Muñoz et al. (2014) while Filioglou et al. (2016) present a sensitivity study of the effect of the algorithm's input parameters to the final profiles. Furthermore, the LIRIC retrievals have already been evaluated for volcanic and desert dust particles by Wagner et al. (2013) showing that the inversion can be accurate for two quite different types of aerosol. The aerosol extinction products of LIRIC

has also been compared against the respective products from the Generalized Aerosol Retrieval from Radiometer and Lidar Combined data (GARRLiC) algorithm and against the retrievals from raman lidar measurements (Bovchaliuk et al., 2016).

## 3 Methodology

The analysis is divided in two parts. Section 3.1 corresponds to the preprocessing of the algorithm's and the model's estimates in order to calculate comparable final products. The characterization procedure of the lidar profiles is also described there. The

methodology of the comparison is included in section 3.2. Two sample cases are presented in section 3.3 aiming to give an example of the products that the algorithm and the models can provide and also to demonstrate typical problems that occur in the analysis.





## 3.1 Preprocessing

In the first part of this section, the preprocessing procedure of the lidar measurements is described. In the second part, we present the methodology that was applied in order to characterize the lidar profiles.

### 3.1.1 Lidar preprocessing

The LIRIC algorithm requires both the raw lidar signal resulting from the atmospheric elastic backscattering in 355nm, 532nm and 1064nm and the Version 2 Inversions from AERONET. Lidar measurements performed in Thessaloniki during the period 2013-2015 were used for this purpose. Before September 2012, the setup of the lidar system in Thessaloniki was lacking a 1064nm channel that is necessary for the LIRIC inversion. A manual cloud screening process was also applied in order to remove all the cloud affected lidar measurements since LIRIC is not designed for cloud layers. In addition, only daytime
measurements were used since the sunphotometer only operates during daytime. It is important to mention that both instruments are located close to one another and at the same altitude above sea level (see section 2.1 and section 2.2).

The sunphotometer data are processed by the AERONET algorithms in order to calculate the necessary aerosol properties which are required as input for the algorithm. In our analysis, the closest AERONET inversion to the central time of the lidar measurement was selected for the LIRIC retrievals. Cases with an absolute time difference that exceeded 3 hours between the
sunphotometer measurement time and the central time of the lidar measurement were excluded. The lidar signal preprocessing is performed directly in LIRIC (Chaikovsky et al., 2016) and includes the averaging, smoothing, background correction and range correction procedures as well as the normalization of the lidar signals and also the selection of the lower and upper height boundaries in the signal where the LIRIC inversion is going to be performed. All the signals are adjusted to a common vertical resolution with a constant step of 15m. A lower height boundary has to be determined due to the overlap function of the lidar
system. In the current dataset the full overlap height was calculated at 900m. The lower boundary is set to 600m where the overlap function is still above 90%. Below this height the lidar signals are considered constant. As it was mention in section 2.3, this can produce uncertainties since the radiometric data corresponds to the whole atmospheric column and most of the aerosol are usually located in the boundary layer, close to the ground. According to Granados-Muñoz et al. (2014) the selection of the lower limit is the main source of error. They estimate the maximum uncertainty due to such intrinsic errors at 33%
and the overall profile error to stay below 15% most of the time. The upper boundary depends on the maximum height where aerosol exist in a significant quantity and can vary depending on the atmospheric conditions. The output data of the algorithm includes vertical volume concentration profiles of the fine and coarse mode particles in ppbv. By summing the concentration in the two modes one can calculate the total aerosol concentration.

The vertical resolution of the LIRIC products and the model products is different so it was necessary to upscale LIRIC to the
resolution of each model. This can affect the vertical structure of the profiles for individual cases but in a statistical analysis those effects will be smoothed (Binietoglou et al., 2015). The temporal resolution of CAMx forecasts is one hour while the temporal resolution of BSC-DREAM8b forecasts is three hours. Each of the retrieval algorithm's profiles were matched to the models' profile closest to the central lidar measurement. Since LIRIC derived concentration values are in ppbv units while





both models' profiles are in $\mu g \cdot m^{-3}$ units, it was necessary to apply a unit conversion that also requires the aerosol density. Despite the aerosol component densities of CAMx being known, we preferred to convert ppbv to $\mu g \cdot m^{-3}$ (equation 1) since $\mu g \cdot m^{-3}$ is more widely used as a concentration unit.

$$c_{\mu g \cdot m^{-3}} = 10^3 \cdot c_{ppbv} \cdot \rho_{g \cdot cm^{-3}} \tag{1}$$

Where $\rho$ is the mean aerosol density. Typical density values of 1.5 and 2.6 $g \cdot cm^{-3}$ for the fine and coarse mode particles were used respectively (Bukowiecki et al., 2011; Schumann et al., 2011; Kokkalis et al., 2013).

### 3.1.2 Characterization of the lidar profiles

It was mentioned in section 2.3 that the emission inventory of CAMx lacks the biomass burning aerosol emissions from wildfires. Additionally, the desert dust emissions are taken into account only indirectly in the CAMx chemical boundary
conditions (section 2.3). In order to examine the effect of those cases on the comparison we group the cases in four categories. The first one is the total of the cases that will be refer to as 'all'. The second one contains the cases identified as biomass burning wildfire aerosol and will be referred to as "fires" from now on. The aerosol characterization is performed using a combination of model simulations and satellite data. It is described in the next paragraph. When the category 'fires' is screened from the category 'all', the category 'non fires' is formed. It contains the continental and desert dust cases. Finally, the desert dust cases
are also isolated and are included in the category 'dust'.

The backward trajectories from HYSPLIT in conjunction with fire pixel data from the MODIS Terra and Aqua Global Monthly Fire Location Product (MCD14ML) are used to identify the fire cases. The dust cases characterization also utilizes the HYSPLIT trajectories in combination with the BSC-DREAM8b profiles. A pair of 6-day back-trajectories, one arriving in the boundary layer region and another in the Free Troposphere, were used. The technique that was utilized in order to estimate
the boundary layer height per case is described in section 3.2. A high fire spot density on a region where the air masses are passing near ground is applied as a criterion for the wildfire cases identification.

The trajectories for the continental, desert dust and biomass burning cases are presented in figure 1 with blue (a and b), orange (c and d) and black color (e and f) respectively. The left column contains the air masses that arrive in the boundary layer, typically around 1km while the right column the ones that arrive in the Free Troposphere, usually ranging between 3 and
4km. Each trajectory is accompanied by the corresponding accumulated 6-day fire pixels.

### 3.2 Comparison Strategy

The first part of the evaluation of CAMx is based on the comparison of the aerosol concentration with the LIRIC estimates for the 'all' category (section 4.1). The effect of the wildfire cases on the results is also examined in this section. In the second part, the accuracy of CAMx in events of transported Saharan dust is investigated (section 4.2).
The concentration profiles of all the fine aerosol components (table 1) are summed to create the fine mode concentration profile of CAMx. The same applies to the coarse aerosol components of table 1. The water content is included in the fine mode since all the hygroscopic particles are fine.



The number of cases that take part in the comparison as well as the high variability between individual cases emphasize the need of statistics in the analysis. The mean profiles of the two models and LIRIC are calculated across the vertical range. The center of mass is also calculated for each case and each mode. It provides additional information on the height where the majority of the particles are located.

The Planetary boundary layer (PBL) marks the top of the layer where the atmosphere is well mixed and the local aerosol component is also expected to be significant. On the other hand, the Free Troposphere (FT) above, is related to much less mixing and a stronger transported aerosol component is expected. Consequently, a comparison between CAMx and LIRIC in the boundary layer and in the free troposphere would be useful in order to investigate the accuracy of the model in these quite different atmospheric regions. Thus, we calculate the fine and coarse integrated mass in the boundary layer and in the Free
Troposphere for each case.

There are multiple techniques in order to estimate the boundary layer from lidar measurements (e.g., Flamant et al., 1997; Menut et al., 1999; Brooks, 2003). Baars et al. (2008) apply a wavelet covariance transform on the lidar range-corrected signal in order to translate signal layers to maxima and minima of the Wavelet Covariance Transform (WCT). Here we apply the transform to the total concentration derived by LIRIC. However this is not enough to automatically identify the PBL since the
boundaries of multiple layers will be retrieved. Identification criteria are necessary for the selection of the PBL height. The top of the layer between 400m and 2.5km with the minimum value in the transformed signal is chosen as the boundary layer height. The wavelet transform is applied to the LIRIC concentration profiles before the upscaling of the resolution.

To perform the integration below and above the PBL a lower and an upper boundary is required. The LIRIC inversion requires a height where the aerosol content is not significant. This height is usually different for each case and typically ranges
between 3km and 9km. On the other hand, CAMx always provides values up to 9.5km while BSC-DREAM8b provides values up to 15km. We used the ground level as the common lower boundary and the upper limit of CAMx (9.5km) as the common upper boundary. Since LIRIC profiles usually end below this upper limit, the last value of each profile is considered constant up to 9.5km.

Metrics for the center of mass and the integrated mass in the boundary layer and in the Free Troposphere are also calculated.
This includes the mean values and the standard deviations for the algorithm and the model, the mean bias error, the mean fractional error the root mean square error (RMSE), the correlation coefficient and the least squares fit slope and axis intersect values. The equations for some of the metrics can be found on table 2.

In order to demonstrate how the comparison strategy was applied we present two distinct cases which includes the aerosol typing procedure, the concentration profiles comparison and the optical products that can be derived by the LIRIC algorithm.

**3.3  Example Cases**

The products of LIRIC, CAMx, BSC-DREAM8b and HYSPLIT for 2 case studies are presented here.

The main product of LIRIC is the fine and coarse mode concentration profiles. Additionally, the aerosol extinction and backscattering efficiencies per mode and per wavelength (355nm, 532nm, 1064nm) are also derived. The total extinction and backscatter coefficient profiles, symboled as $a$ and $b$ respectively, for the three wavelengths are calculated using the equations





and 3, were $Q_{ext}$ is the extinction efficiency and $Q_{bsc}$ is the backscattering efficiency. The concentration for the fine and coarse mode is marked as $c_f$ and $c_c$ respectively.

$$a(\lambda, z) = Q_{ext,f}(\lambda) \cdot c_f(z) + Q_{ext,c}(\lambda) \cdot c_c(z) \tag{2}$$

$$b(\lambda, z) = Q_{bsc,f}(\lambda) \cdot c_f(z) + Q_{bsc,c}(\lambda) \cdot c_c(z) \tag{3}$$

Then, from the LIRIC estimated extinction and backscatter profiles, the lidar ratio and the angstrom exponent can also be calculated. Furthermore, as it was mentioned in section 3.1, the concentration values below 600m are kept constant in the LIRIC inversion and this also applies to the optical products.

   A typical continental case on the 13th of January 2014 and a typical dust case on the 27th of August 2013 were selected in
order to demonstrate the comparison results for two quite different typical aerosol types. The continental case is demonstrated on the left of figure 2 and the Saharan dust case on the right column.

   The trajectories indicate a continental case on 13th of January 2014. They are presented in figure 2a. Additionally there could be some mixing with marine particles from the Adriatic sea. The air mass arriving in the free troposphere should be clean according to HYSPLIT since the trajectory is elevated, always above 3km. The concentration profiles of LIRIC and
CAMx are presented in figure 2c and figure 2d. According to LIRIC the fine mode is dominant below 1km. Above that height there is a coarse dominant layer which could be the result of mixing with marine aerosol. The fine mode aerosol concentration profile of CAMx seems in good agreement with LIRIC in the near range, below 2km. On the other hand, CAMx fails to predict the coarse mode. Above 6km, an overestimation of CAMx can be observed in both profiles.

   The four optical products are also presented. The upper part of figure 2g panel contains the aerosol backscatter and extinction
coefficient profiles and the lower part the lidar ratio and the Angstrom exponent profiles. The lidar ratio values are around 70 $\text{sr}^{-1}$ at 355nm and 65 $\text{sr}^{-1}$ at 532nm below 1km. The Angstrom values are near 1.6 for the 355-532nm exponent and at 2.1 for the 532-1064nm exponent. Giannakaki et al. (2010) report a mean lidar ratio at 355nm of $56 \pm 23$ $\text{sr}^{-1}$ and a backscatter-related Angstrom exponent at 355-532nm of $1.4 \pm 1.0$ for the continental polluted aerosol class in Thessaloniki. Between 1km and 2km the lidar ratio at 355nm drops to 55 $\text{sr}^{-1}$ and the lidar ratio 532nm at 45 $\text{sr}^{-1}$. The Angstrom exponent in both
regions drops near 1.3. According to Giannakaki et al. (2010) those values are still within the range of the acceptable range of the continental polluted class.

   As far as the second case is considered, the trajectories indicate an event of transported Saharan dust in the FT (figure 2b). The air masses that arrive in the PBL seem to contain marine aerosol from the Mediterranean, probably mixed with emissions from local urban sources. A strong coarse mode can be observed in figure 2e both for the layer below 1.5km and the layer
above. Despite CAMx predicting a promising fine mode (figure 2d), it fails to do so for the coarse particles. This is the reason why the BSC-DREAM8b concentration profile is presented here instead. This model describes the dust layer between 2km and 5km much better. Below 2km the model isn't compatible any more with the observation.

   As far as the optical products are concerned, the lidar ratio at 355nm and 532nm values range between 40 and 50 $\text{sr}^{-1}$ for the whole profiles while the Angstrom exponent ranges between 1.0 and 1.5 for both spectral regions (figure 2h). Giannakaki





et al. (2010) calculate a lidar ratio at 355nm of $52 \pm 18$ sr$^{-1}$ and a backscatter-related Angstrom exponent at 355-532nm of $1.5 \pm 1.0$ for the Saharan dust aerosol class at Thessaloniki which seems compatible with this case. In the next section the statistical analysis is presented and the results are discussed.

## 4  Discussion

An ensemble of 24 measurements, that fulfill the criteria described in section 3.2, took part in this comparison. These cases constitute the category 'all'. In the first part of this section (section 4.1) the comparison between LIRIC and CAMx is presented. In the second part (section 4.2) the accuracy of CAMx in case of Saharan dust events is investigated using the simulations of BSC-DREAM8b.

### 4.1  Comparison between LIRIC and CAMx

In this section the simulated profiles of CAMx are compared against the observational profiles of LIRIC. The vertical mean profiles derived from LIRIC and CAMx for the fine and coarse mode particles are displayed in figure 3a and figure 3b. The solid lines correspond to the 'all' category, while the dashed lines correspond to the 'no fires' category, which consists of 17 measurements. It can be seen that the fine mode mean concentration profiles show a good agreement between LIRIC and CAMx. The vertical structure also seems to bear some similarities. There is an overall slight overestimation by the model which becomes more significant above 2km. Removing the 7 wildfire cases modifies the LIRIC mean profile towards slightly lower values and the CAMx mean profile towards slightly higher values, leading to smaller discrepancies below 1km. Details on the behavior of the model in the boundary layer and the free troposphere for both the 'all' and the 'no fires' categories can be found in the next paragraphs. As far as the coarse mode is considered, it seems to be severely underestimated by the model especially below 3km. Above 4.5km the model seems to overestimate the aerosol concentration. The reasons of these discrepancies could be connected with the model's emissions inventory and the boundary conditions. Some insight can be offered by individually inspecting the contribution of each aerosol component in the final product.

The mean concentration profiles of the aerosol components that consist the fine and coarse mode of CAMx are presented in figure 3c and figure 3d. The aerosol components of table 1 are grouped into categories that follow the OPAC formalism (Hess et al., 1998). The Particulate Chloride and Sodium components form the "sea salt" category. The Aerosol Water Content, the Primary Elemental Carbon, the Fine Crustal and Coarse Crustal as well as the Fine Other Primary and Coarse Other Primary components are independent and form the categories "water", "soot", "soil fine", "soil coarse", "other fine" and "other coarse" respectively. The particulate Nitrate, the Sulfate and the particulate Ammonium are all hygroscopic and are grouped into the "water soluble" category. The rest of the species are all organic and are grouped in the "organic insoluble" category. The Fine Other Primary component contains the fine particles that are considered as inert by the model as well as a part of the fine sea salt that cannot be categorized neither as Particulate Chloride nor as Sodium. The Coarse Other Primary component contains the coarse particles that take part in chemical reactions like nitrate, sulfate, ammoniac, black carbon and primary organic aerosol, as well as the coarse sea salt and all the other coarse particles that are considered as inert by the model.





The dominant component below 2km is the water soluble one, followed by the fine sea salt component and the water content. Above 2km the other fine component becomes significantly stronger with a maximum at 3-4km. This region matches with the one where CAMx overestimates the fine mode. Consequently, the "other fine" component could be connected to these biases. As far as the coarse mode is considered, the majority of particles belong to the soil coarse component. Both components seem responsible for the overestimation above 4km in the coarse mean profiles since the highest values in both both component profiles are located above 2km. In order to investigate if only one of them or both are also responsible for the large bias below 3km we isolate the dust component by selecting only the dust cases. This comparison is presented in section 4.2. Below, the center of mass and the integrated mass are calculated in order to further quantify the comparison results.

The center of mass (table 2) provides information on the height where most of the particles are located. It is presented in LIRIC against CAMx center of mass scatter plots (figure 4) for the fine and the coarse mode. The least squares fit line and the correlation coefficient for the 'no fires' category is displayed in the plots. A synopsis of the center of mass metrics can be seen in table 3. The 'no fires' category is not included for the coarse aerosol. LIRIC estimates a mean center of mass value of $1.06 \pm 0.23$ km for the "all" category which increases slightly for the "no fires" category in the fine mode. CAMx predicts a mean center of mass at $1.36 \pm 0.47$ km that doesn't change much if the wildfire cases are excluded. Thus, the resulting mean bias is 0.30 km and the fractional bias is 24.8% and they improve to 0.20km and 16.9% respectively. The root mean square error (RMSE) stays constant near 0.55km. Consequently, the height where most of the fine particles are located seems well-predicted by the model. As far as the coarse mode is considered, LIRIC gives a mean value of $1.36 \pm 0.60$ km while CAMx predicts the center of mass at $3.16 \pm 1.21$ km. As a result, the mean bias and the RMSE are much larger here, which is expected given the large discrepancies that were discussed in the previous paragraph. The correlation coefficient for the fine mode is 0.20 and increases to 0.34 when the wildfire cases are excluded. While the center of mass is useful when examining the vertical distribution and the location of the maximum concentration, it doesn't provide any insight on the concentration itself. Additional information on the accumulated concentration within an atmospheric region can be provided by calculating the integrated mass.

The comparison of the integrated mass in the boundary layer and the free troposphere is displayed in LIRIC against CAMx integrated mass scatterplots (figure 5) and the metrics are presented in table 4. The coarse mode isn't included since it seems to be significantly biased by the model.

The behavior of the model in the boundary layer is examined first. LIRIC estimates a mean integrated mass at $18.8 \pm 16.0$ $mg \cdot m^{-2}$ against a CAMx derived value of $14.1 \pm 10.5$ $mg \cdot m^{-2}$ for the "all" category which change to $14.1 \pm 8.6$ $mg \cdot m^{-2}$ and $16.1 \pm 10.5$ $mg \cdot m^{-2}$ respectively for the "no fires" category. The mean bias changes from -4.7 $mg \cdot m^{-2}$ to 2.0 $mg \cdot m^{-2}$ that translates to a fractional bias shift from -28.6 to 13.2%. The RMSE improves from 14.7 $mg \cdot m^{-2}$ to 6.1 $mg \cdot m^{-2}$. The correlation coefficient is at 0.44 and the least square fit slope at 0.29 for the "all" category but they both improve to 0.81 and 0.99 respectively when the wildfire cases are removed. The results that occur for the "no fires" category indicate that, in the boundary layer, the lack of wildfire emissions in CAMx shouldn't be neglected as it obviously affect the performance of the model.





The behavior of the model in the free troposphere is quite different. The LIRIC mean value is $18.8 \pm 16.0 \, \mathrm{mg \cdot m^{-2}}$ against a CAMx mean value of $14.1 \pm 10.5 \, \mathrm{mg \cdot m^{-2}}$. By excluding the wild fire cases the LIRIC mean stays almost the same, at 23.1 $\pm$ $18.4 \, \mathrm{mg \cdot m^{-2}}$ but the CAMx mean increases to $32.5 \pm 31.9 \, \mathrm{mg \cdot m^{-2}}$. This causes the mean bias and RMSE to actually increase from $8.1 \, \mathrm{mg \cdot m^{-2}}$ and $33.3 \, \mathrm{mg \cdot m^{-2}}$ to $16.6 \, \mathrm{mg \cdot m^{-2}}$ and $35.6 \, \mathrm{mg \cdot m^{-2}}$ respectively. The fractional bias is also

negatively affected and increases from 28.5 to 53.2%. On the other hand, the correlation coefficient is at 0.25 and the least square fit slope at 0.45 for the "all" category and they improve to 0.36 and 0.69 respectively for the "no fires" category. All things considered, the removal of the wildfire cases doesn't seem to positively affect the agreement between the CAMx and LIRIC in the free troposphere. This is also depicted in the scatterplot (figure 5, on the right) where the "fires" cases (red) are all quite close to the unity line in contrast with the boundary layer (figure 5, on the left) where most of those cases seem to be

outliers. Discrepancies in the free troposphere could also be attributed to the fact that the LIRIC profile above the height where the aerosol load is insignificant is considered constant (see sections 3.1.1 and 3.2) while the CAMx profile is still active in that region. Additionally, a possible preference of the biomass burning layers to arrive in the PBL over Thessaloniki could also be associated with this behavior.

Taking into account the performance of CAMx in both atmospheric regions, it appears to predict the fine aerosol more accu-

rately in the boundary layer than in the free troposphere. This behavior seems consistent with the previous results considering that CAMx tends to estimate higher concentration values in higher altitudes (figure 3 and figure 4) in the fine mode, which could also be attributed to the "other fine" component. When the wildfire cases are not taken into account the performance of the model improves but only in the PBL. Possible causes for the discrepancies between LIRIC and CAMx could be the aerosol emission inventory of CAMx and the chemical boundary conditions. We have shown here that cases that also include wildfire

aerosols are a challenge for the model since those emissions are not included at all. It has been stated in section 2 that the soil dust resuspension emissions are also not included and that the Saharan dust emissions are included indirectly. The large discrepancies in the coarse mode mean profile could be connected to a combination of the lack of dust resuspension emissions and of biased Saharan dust boundary conditions. In the next section the behavior of CAMx in transported dust events is analyzed.

### 4.2 Dust cases analysis using BSC-DREAM8b

In this section the coarse mode products derived by LIRIC are compared against the simulations of both CAMx and BSC-DREAM8b. Out of the initial dataset of 24 measurements, 6 were identified as dust cases. We have to mention that the focus of this study is not a validation of BSC-DREAM8b since the number of dust cases wouldn't be sufficient. The BSC-DREAM8b model is used here solely to support the analysis of CAMx. That being said, we aim to isolate the coarse desert dust component and compare between LIRIC and each model in order to investigate if the observed large discrepancies in the coarse mode of

CAMx are also present in the simulations of a model that specializes in desert dust. While it is feasible to get the coarse dust profile with CAMx from the coarse profile by selecting only the "soil coarse" component (table 1) this is not the case for the coarse profile of LIRIC. Having applied the aerosol characterization of section 3.1.2, it is reasonable to assume that the coarse mode of the selected dust cases is almost entirely attributed to dust. Binietoglou et al. (2015) also use a dataset of measurements that were flagged as desert dust in order to compare between the observations and the simulations of the dust transportation



models. Obtaining the coarse dust of BSC-DREAM8b is also not an option because this model provides total dust profiles. d'Almeida (1987) mention that the contribution of the fine dust should be low, especially near where is emitted. Mamouri and Ansmann (2014) have separated the fine and coarse mode dust profiles during an outbreak event. They report that while the fine dust contribution in the mass concentration was significant near the ground in that transported dust case, it stayed well

below 20% of the total concentration above 400m. Consequently, the use of the total dust profile of BSC-DREAM8b shouldn't jeopardize much the validity of the comparison with LIRIC, especially in the free troposphere.

The mean profiles of the aerosol mass concentration in the coarse mode is presented in figure 6. The comparison between CAMx and LIRIC can be seen on the left while the comparison between BSC-DREAM8b and LIRIC is presented on the right. It is obvious that CAMx underestimates the concentration by providing values that never exceed $10\,\mu g \cdot m^{-3}$ while the LIRIC

mean values raise up to $45\,\mu g \cdot m^{-3}$ and potentially much higher for selective cases. On the other hand, BSC-DREAM8b values are close to the ones derived by LIRIC between 2km and 4km, ranging between 20 and $40\,\mu g \cdot m^{-3}$. Below 2km, even BSC-DREAM8b seems to underestimate the concentration. This could be attributed to mixing with coarse particles other than desert dust in this region. This scenario is further supported by taking into account the "dust" category trajectories that arrive in the PBL (figure 1c) in section 3.3. In the next paragraphs the center of mass and the integrated mass comparison is presented

in a way similar to section 4.1.

The center of mass comparison between LIRIC and BSC-DREAM8b is presented in figure 7a accompanied by the observational and modeled metrics in table 5. CAMx predicts a center of mass at $2.55 \pm 1.13$ km which is actually close to the value of BSC-DREAM8b at $2.38 \pm 0.57$. Discrepancies in the mean bias here originate mainly from differences in the LIRIC center of mass product that occur due to differences from the interpolation of the LIRIC profile to the vertical resolution of each model.

The correlation coefficient and the least square fit slope values are at -0.13 and -0.28 respectively between the algorithm and the air quality model but they improve at 0.44 and 0.39 when the dust transportation model is used instead. Binietoglou et al. (2015) estimate a correlation coefficient of 0.38 for the same model and for a dataset of 69 dust profiles.

As far as the integrated mass is considered, the comparison of LIRIC vs BSC-DREAM8b is also presented in figure 7, the boundary layer in figure 7b and the free tropospheric region in figure 7c. The agreement is best in the free tropospheric region

for the dust specialized model which is in accordance with the profiles of figure 6. The mean values for BSC-DREAM8b in the FT are $90.1 \pm 94.9\ mg \cdot m^{-2}$. They are quite close to the mean values of LIRIC at $108.9 \pm 72.9\ mg \cdot m^{-2}$ resulting to the lowest absolute mean bias of table 6 at $-18.8\ mg \cdot m^{-2}$ and a fractional error of -18.9%. The respective fractional error of CAMx in the FT is -103.3% . The correlation between LIRIC and BSC-DREAM8b in the same region is high, at 0.91, and the least square fit slope is 1.19 in contrast with the PBL region where the slope is much lower at 0.23 despite the correlation being

similar. Binietoglou et al. (2015) report a correlation of 0.54 for the integrated dust concentration in the whole profile for their dataset of 69 measurements. As it has already been mentioned, discrepancies between the algorithm and the models can occur from the non dust aerosol component. This can be crucial for the PBL region, where particles from various sources are mixed.

The comparison between LIRIC and the two models resulted to a much better performance of BSC-DREAM8b than CAMx in the free troposphere for the dust cases. The dust specialized model was able to reproduce values similar to the observations,

leading to the conclusion that the "soil coarse" component of CAMx is definitely underestimated. This behavior could be linked





to the model's lack of direct Saharan dust emissions within the domain (see section 2.3). Having said that, the desert dust cases are not the majority and by removing the 6 dust cases out of the 24 total cases didn't improve much the coarse profile of CAMx. In order to explain these large biases (section 4.1) we have to assume that the "other coarse" component is also biased, most probably underestimated below 4km. This component contains the coarse sea salt and also the coarse ammoniac, sulfate and

nitrate particles (see table 1 and section 4.1) that are all supposed to be hygroscopic. Despite that, the "other coarse" component is not treated as hygroscopic by the model. Thus the water absorption and the hygroscopic growth of the coarse particles is not taken into account at all. Consequently, the absence of any water content in the coarse mode of CAMx could lead to an underestimation of the model since LIRIC concentration profiles include the water content absorbed by the fine and coarse particles.

**5  Conclusions**

The evaluation of CAMx resulted in a mean profile in the same order of magnitude and of similar vertical distribution with the observational one for the fine mode. The mean center of mass of the model is different by only 0.3km from the respective value of the algorithm which translates to a mean fractional bias of 24.8%. The correlation coefficient is estimated at 0.20 and improves at 0.34 when the wildfires are excluded. The integrated mass comparison indicates a better performance of the model

in the boundary layer than in the free troposphere. For the "no fires" category, the mean fractional bias improves from -28.6 to 13.2%. At the same time, the correlation coefficient of the integrated mass rises from 0.44 to 0.86 and the least squares fit slope from 0.29 to 0.99. The comparison in the free tropospheric region, on the other hand, is not clearly benefited from the removal of those cases. All things considered, there are strong indications that the lack of the wildfire emissions in CAMx affect it's performance concerning the mass concentration of the fine mode particles that arrive in the boundary layer.

The coarse mode mean profile of CAMx, on the other hand seems to be greatly underestimated below 4km and overestimated above. Consequently, the vertical structure is also incompatible with the observational data. The behavior of the "soil coarse" component of CAMx was tested using selected dust cases and the desert dust dispersion model BSC-DREAM8b. Both models underestimate the concentration in the boundary layer. In the free troposphere, BSC-DREAM8b achieves a fractional bias of -18.9% and a correlation coefficient at 0.91 in contrast with CAMx where the same metrics are estimated at -103.3% and

-0.69 respectively. The center of mass is also better correlated in the dust model. Care has to be taken, however, because the interpolation of the LIRIC profiles in the models' vertical resolution affects the center of mass and shifts the LIRIC mean value by from 1.75km for the resolution of CAMx to 2.14km for the resolution of BSC-DREAM8b. Since BSC-DREAM8b outperforms CAMx, at least in the free troposphere, it is reasonable to assume that the "soil coarse" component is a source of bias. Furthermore, the fact that the small number of the dust cases (6 out of 24) is not enough to explain the large discrepancies

in the coarse mode mode between LIRIC and CAMx, it is likely that the "other coarse" component is also biased. This could be linked to the fact that it consists of many subcomponents, some of which in theory are hygroscopic, like the coarse sea salt, but they are not treated as such by the model, possibly leading to underestimations in the aerosol concentration.



This study shows that the LIRIC code, based on the synergy of and lidar measurements can be used in order to evaluate an air quality model like CAMx as far as the aerosol mass concentration is considered. Furthermore, models specialized in particular types of emissions, like the BSC-DREAM8b dust transportation model, can be used along with LIRIC in order to help isolate one specific aerosol component that the air quality models provides or completely lacks. That way, the components

can be tested individually, making it possible to directly associate biases with a specific type of emissions. Here, for example, we concluded that the lack of a wildfire component, the desert dust component and the remaining coarse component are all potential sources of bias in the modeled aerosol concentration profiles. The emissions that are associated with these aerosol types can then be examined and proper corrections could be applied in order to improve the overall performance of the model. Finally, if such comparisons are successful then the simulations of the model can also be utilized in the aerosol classification

procedure of the lidar measurements since the individual aerosol components of the model could provide insight on the origin of the main aerosol layers.

*Acknowledgements.*  The authors would like to acknowledge the EU projects MACC-III (Monitoring Atmospheric Composition and Climate - III, Grant agreement no: 633080) and MACC-II project (Monitoring Atmospheric Composition and Climate - Interim Implementation, Grant agreement no: 283576). The simulated results presented in this research paper have been produced using the EGI and HellasGrid

infrastructures. The ACTRIS-2 project from the European Union's Horizon 2020 research and innovation programme under grant agreement No 654109 is gratefully acknowledged. The authors would also like to acknowledge the support provided by the Scientific Computing Center at the Aristotle University of Thessaloniki throughout the progress of the work on air quality forecasting. BSC-DREAM8b simulations were performed on the Mare Nostrum supercomputer hosted by Barcelona Supercomputing Center-Centro Nacional de Supercomputación (BSC-CNS). S. Basart wants to acknowledge the CICYT project (CGL2013-46736). Elina Giannakaki acknowledges the support of the Academy

of Finland (project no. 270108).



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





**Table 1.** CAMx Aerosol Components Synopsis. The Fine Other Primary is mainly consisted out of the fine aerosols that are treated as inert by the model. It also contains small part of the fine sea salt that cannot be treated as Particulate Chloride or as Sodium. The Coarse Other Primary component includes all the coarse aerosols that are not crustal (nitrate, sulphate, ammoniac, black carbon and primary organic aerosols) as well as the coarse sea salt and all the other particles that are considered as inert by the model.

| Components | Hygroscopic | Mode | Category |
|---|---|---|---|
| Particulate Nitrate (NO3) | yes | fine | water soluble |
| Sulfate (SO4) | yes | fine | water soluble |
| Particulate Amonium (NH4) | yes | fine | water soluble |
| Aerosol Water Content (H2O) | – | fine | water |
| Anthropogenic SOA* | no | fine | organic insoluble |
| Biogenic SOA* | no | fine | organic insoluble |
| Polymerized Anthropogenic SOA* | no | fine | organic insoluble |
| Polymerized Biogenic SOA* | no | fine | organic insoluble |
| Sodium (Na) | yes | fine | sea salt fine |
| Particulate Chloride (Cl) | yes | fine | sea salt fine |
| Primary Organic Aerosol | no | fine | organic insoluble |
| Primary Elemental Carbon (C) | no | fine | soot |
| Fine Other Primary | no | fine | other fine |
| Fine Crustal | no | fine | soil fine |
| Coarse Other Primary | no | coarse | other coarse |
| Coarse Crustal | no | coarse | soil coarse |

*SOA = Secondary Organic Aerosol





**Table 2.** Common Metrics Equations. The model data is defined as M and the lidar observational data as O. The a and c symbols correspond to the height and the concentration respectively.

| Metric | Equation |
|---|---|
| Center of Mass | $\dfrac{\int z \cdot c \cdot dz}{\int c \cdot dz}$ |
| Mean Bias | $\dfrac{1}{N} \sum\limits_{i=1}^{N} (M_i - O_i)$ |
| Mean Fractional Bias (%) | $\dfrac{200}{N} \sum\limits_{i=1}^{N} \dfrac{(M_i - O_i)}{(M_i + O_i)}$ |
| RMSE | $\left[ \dfrac{1}{N} \sum\limits_{i=1}^{N} (M_i - O_i)^2 \right]^{\frac{1}{2}}$ |





**Table 3.** Center of Mass Metrics. The category 'All' corresponds to the total of the cases (24 in total) while the category 'No Fires' refers to the cases that are not classified as wildfires (17 out of 24). The mean, standard deviation, mean bias and root mean square error values are in [**km**] units. The Pearson's correlation coefficient (r) and the least square fit slope (a) and intercept (b) values are also calculated.

| Mode | Category | LIRIC Mean | CAMx Mean | Mean Bias | Frac. Bias (%) | RMSE | r | a | b |
|---|---|---|---|---|---|---|---|---|---|
| Fine | All | $1.06 \pm 0.23$ | $1.36 \pm 0.47$ | 0.30 | 24.8 | 0.56 | 0.20 | 0.23 | 1.12 |
| Fine | No Fires | $1.14 \pm 0.44$ | $1.35 \pm 0.50$ | 0.21 | 16.9 | 0.54 | 0.34 | 0.39 | 0.90 |
| Coarse | All | $1.36 \pm 0.60$ | $3.16 \pm 1.21$ | 1.80 | 79.6 | 1.31 | 0.07 | 0.13 | 2.97 |





**Table 4.** Fine Mode Integrated Mass Metrics. Two atmospheric regions are provided. The region below the boundary layer is symbolized as PBL while the region above it, in the free troposphere, is defined as FT. The category 'All' corresponds to the total of the cases (24 in total) while the category 'No Fires' refers to the cases that are not classified as wildfires (17 out of 24). The mean, standard deviation, mean bias and root mean square error values are in $[\mathbf{mg \cdot m^{-2}}]$ units. The Pearson's correlation coefficient (r) and the least square fit slope (a) and intercept (b) values are also calculated.

| Region | Category | LIRIC Mean | CAMx Mean | Mean Bias | Frac. Bias (%) | RMSE | r | a | b |
|--------|----------|------------|-----------|-----------|----------------|------|------|------|------|
| PBL | All | $18.8 \pm 16.0$ | $14.1 \pm 10.5$ | -4.7 | -28.6 | 14.7 | 0.44 | 0.29 | 8.61 |
| PBL | No Fires | $14.1 \pm 8.6$ | $16.1 \pm 10.5$ | 2.0 | 13.2 | 6.1 | 0.81 | 0.99 | 2.16 |
| FT | All | $24.4 \pm 17.4$ | $32.5 \pm 31.9$ | 8.1 | 28.5 | 33.3 | 0.25 | 0.45 | 21.43 |
| FT | No Fires | $23.2 \pm 18.4$ | $40.0 \pm 35.4$ | 16.8 | 53.2 | 35.6 | 0.36 | 0.69 | 24.01 |





**Table 5.** Dust Center of Mass Metrics. Metrics for the two models and LIRIC are provided. A total of 6 out of 24 cases are included in the category 'dust'. Differences in the LIRIC values per model are attributed to the interpolation in different vertical resolution depending on the model. The mean, standard deviation, mean bias and root mean square error values are in [**km**] units. The Pearson's correlation coefficient (r) and the least square fit slope (a) and intercept (b) values are also calculated.

| Model | LIRIC Mean | Model Mean | Mean Bias | Frac. Bias (%) | RMSE | r | a | b |
|---|---|---|---|---|---|---|---|---|
| CAMx | $1.75 \pm 0.51$ | $2.55 \pm 1.13$ | 0.79 | 36.7 | 1.30 | -0.13 | -0.28 | 3.04 |
| BSC-DREAM8b | $2.14 \pm 0.63$ | $2.38 \pm 0.57$ | 0.23 | 10.2 | 0.64 | 0.44 | 0.39 | 1.53 |





**Table 6.** Dust Integrated Mass Metrics. Metrics for the two models in two atmospheric regions are provided. The region below the boundary layer is symbolized as PBL while the region above it, in the free troposphere, is defined as FT. A total of 6 out of 24 cases are included in the category 'dust'. The mean, standard deviation, mean bias and root mean square error values are in $[\mathbf{mg \cdot m^{-2}}]$ units. The Pearson's correlation coefficient (r) and the least square fit slope (a) and intercept (b) values are also calculated.

| Model | Region | LIRIC Mean | Model Mean | Mean Bias | Frac. Bias (%) | RMSE | r | a | b |
|---|---|---|---|---|---|---|---|---|---|
| CAMx | PBL | $39.7 \pm 50.9$ | $3.3 \pm 2.2$ | -36.4 | -169.3 | 50.2 | 0.31 | 0.01 | 2.78 |
| BSC-DREAM8b | PBL | $43.8 \pm 52.6$ | $10.8 \pm 13.2$ | -33.0 | -120.9 | 40.6 | 0.93 | 0.23 | 0.58 |
| CAMx | FT | $106.0 \pm 74.9$ | $33.8 \pm 23.8$ | -72.2 | -103.3 | 92.9 | -0.69 | -0.22 | 57.00 |
| BSC-DREAM8b | FT | $108.9 \pm 72.9$ | $90.1 \pm 94.9$ | -18.8 | -18.9 | 41.3 | 0.91 | 1.19 | -39.27 |



**6 Days Backward Trajectories over Thessaloniki**

**Figure 1.** HYSPLIT 6-day backward trajectories and MODIS fire pixels for the continental (a and b), desert dust (c and d) and biomass burning (e and f) cases. The left column includes trajectories that arrive in the boundary layer while the right column includes trajectories arriving in the free troposphere.






**Figure 2.** Two sample cases are presented here. The left column corresponds to a continental aerosol case on the 13th of January 2014 while the right column corresponds to a dust aerosol case on the 27 of August 2013. The wind back-trajectories (a and b), the fine and coarse mode concentration profiles of liric (c, d, e and f), and the respective optical products (g and h) are include.







**Figure 3. a and b**: Comparison of the mean fine mode (a) and coarse mode (b) aerosol concentration profiles between LIRIC and CAMx. The shaded regions correspond to one $\sigma$ of each average profile.

**c and d**: Presentation of the mean concentration profiles per aerosol component out of which the model's fine mode (c) and coarse mode (d) are consisted (see table 1).





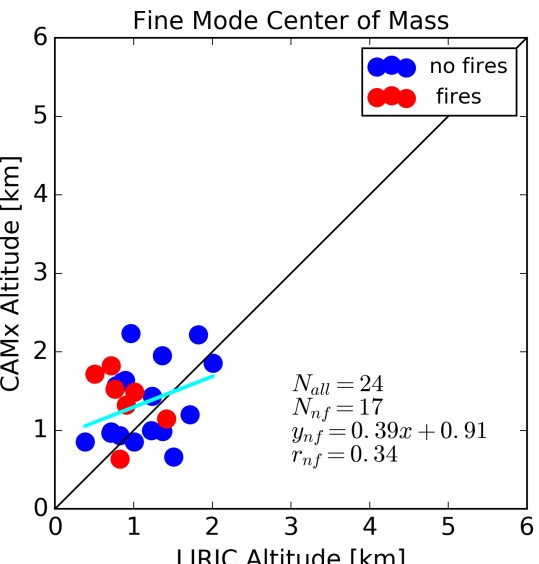
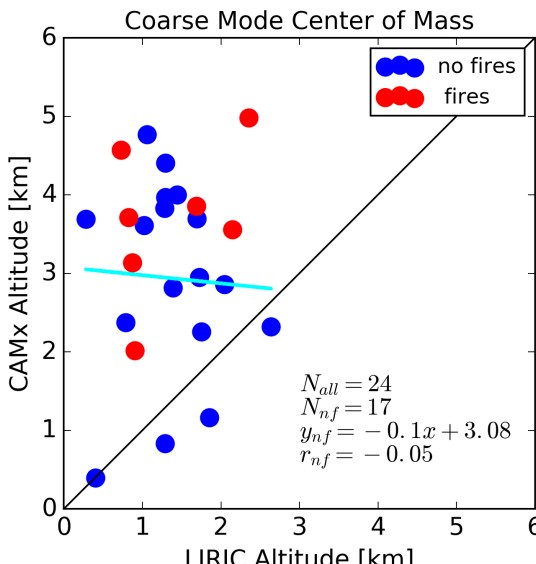

**Figure 4.** Scatterplots of the fine (left) and coarse mode (right) center of mass for LIRIC and CAMx. The biomass burning cases are marked with a red color. The 'nf' label corresponds to metrics of the category 'no fires'. The least square fit line corresponding to the screened cases (blue) is also included.





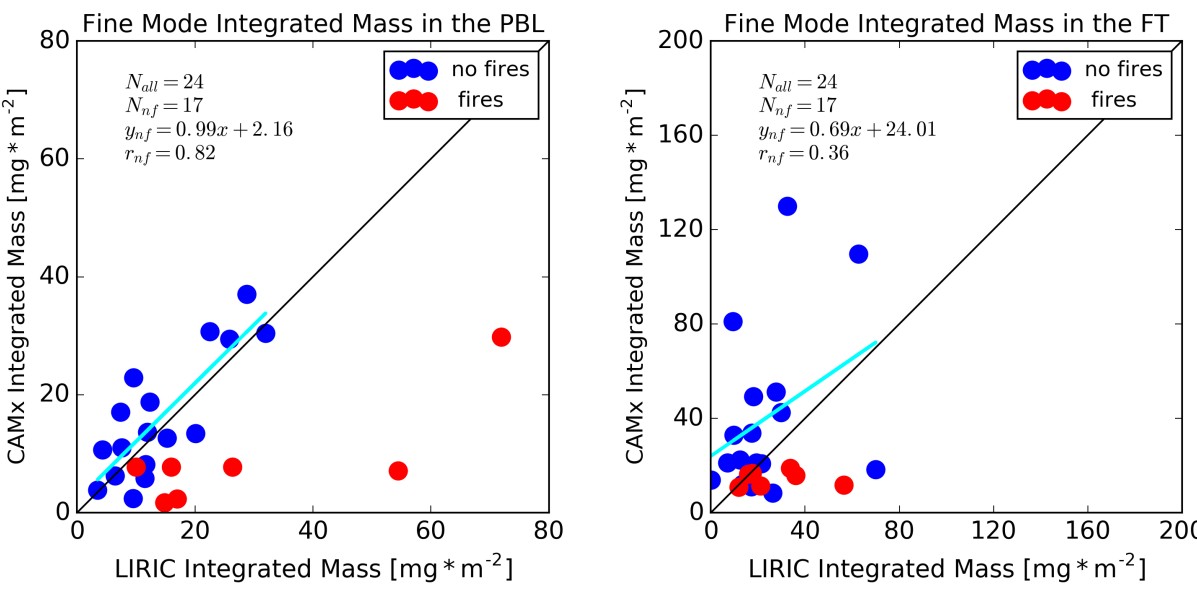

**Figure 5.** Scatterplots of the fine mode integrated mass in the boundary layer (left) and in the free troposphere (right) for LIRIC and CAMx. The biomass burning cases are marked with a red color. The 'nf' label corresponds to metrics of the category 'no fires'. The least square fit line corresponding to the screened cases (blue) is also included.





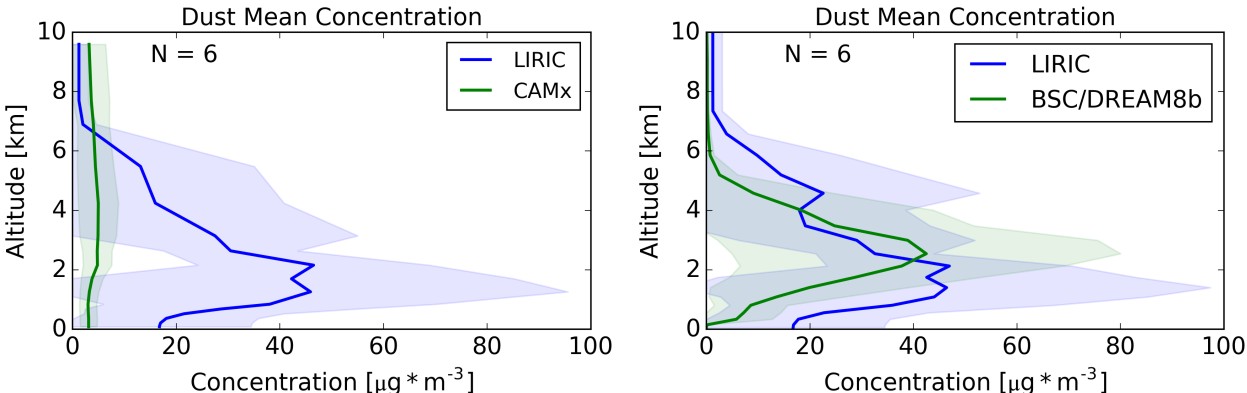

**Figure 6.** Comparison of CAMX (left) and BSC-DREAM8b (right) mean dust concentration profiles with LIRIC. The mean coarse mode profile from LIRIC, the mean soil coarse component profile from CAMx and the mean total dust profile from BSC-DREAM8b are presented. The shaded regions correspond to one $\sigma$ of each average profile. Differences between the LIRIC profiles are attributed to the interpolation of LIRIC in the resolution of each model.





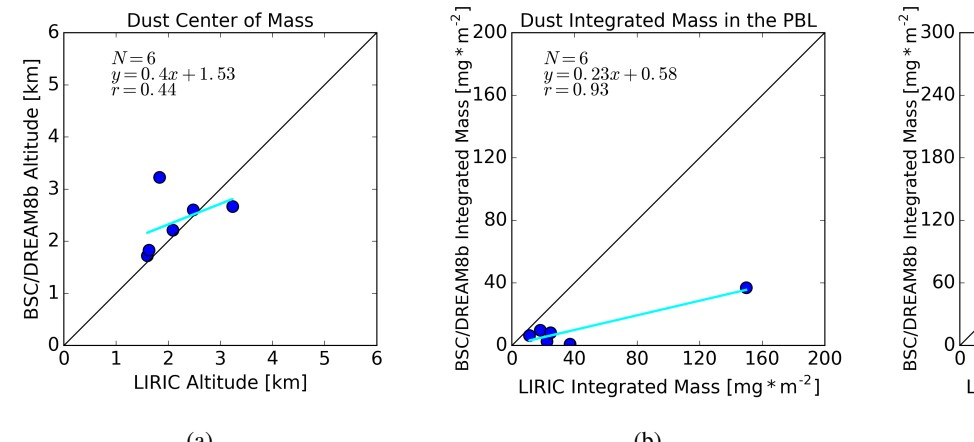

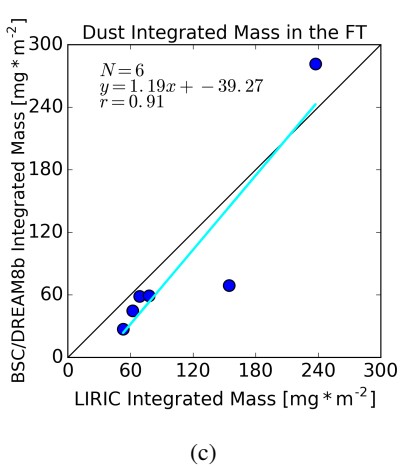

(a)                                    (b)                                    (c)

**Figure 7. a**: Scatterplots of the center of mass of the dust cases for BSC-DREAM8b.

**b**: Scatterplots of the integrated mass of the dust cases for BSC-DREAM8b in the boundary layer.

**c**: Scatterplots of the integrated mass of the dust cases for BSC-DREAM8b in the free troposphere.

The least square fit line is also included. The correlation coefficient for those cases is provided as well.