# Peer review of "Investigating the quality of modeled aerosol profiles based on combined lidar and sunphotometer data"

_Atmospheric Chemistry and Physics, 2016_

## Referee Comment (RC1) · Anonymous Referee #2 · 8 Feb 2017

General comments: An evaluation study is presented to assess the capability of the air quality model CAMx to describe the aerosol conditions over Thessaloniki. The model simulations are compared to combined sun photometer and lidar observations. Backward trajectories and results of a sophisticated dust forecast model are used to attribute shortcomings to a poor representation of biomass burning and desert dust aerosol.

In principle, I like the idea of using different tools, not only measurements, to evaluate the simulations of a specific model and track down shortcomings to suggest model improvements. The evaluation is properly done, although the focus on the comparison with LIRIC data from Thessaloniki only may be too one-sided. Maybe other observa-

tions could be additionally included to underpin the findings.

However, my essential criticism is that the CAMx model is evaluated regarding two aerosol types, which, by design, are not directly computed or only poorly represented. Biomass burning emissions are highly variable in time and space. The actual pollution will largely depend on specific events. Of course, it is not to be expected that the TNO emission database from 2007 in detail is representative for the fire emissions in 2013 – 2015. The same holds for Saharan dust that is not online computed based on modelled winds but input as boundary condition. This must be considered when evaluating the model results, and the conclusions have to be revised in this regard. How exactly is the CAMx model suggested to be improved with this in mind, and based on the evaluation results?

Specific comments: 1. Page 4, line 10: A plot showing the model domains would be very helpful, in particular, to show if relevant Saharan dust sources are included. 2. Page 11, lines 5 – 8: Here and later in Section 4, the study period 2013 – 2015 should be mentioned in order to clearly separate example cases from the broader statistical analysis. 3. Figures 1 and 3 – 7: Please indicate in each figure caption whether the results refer to a specific case or the entire period 2013 – 2015.

---

## Referee Comment (RC2) · Anonymous Referee #1 · 13 Feb 2017

The paper of Siomos et al. presents an interesting example of aerosol model evaluation based on remote sensing observations. The manuscripts highlights the potential and pitfalls for such a comparison, therefore it could be of interest for the wider atmospheric community. The manuscript is worth publishing after addressing several comments listed below.

General comments

1) A main issue with the presented analysis is that the authors compare fine and coarse particles defined in two fundamentally different ways. According the the text, the model's fine mode is defined as particles with aerodynamic diameter less than 2.5um, while LIRIC's fine mode is defined as particles with (optical) diameter less than ∼0.4

– 1.2 um. Before this study is published, the authors should thoroughly discuss this issue and justify why their comparison gives any meaningful results.

2) Desert dust is included in the model only as a boundary conditions and this explains, according the the authors, the poor performance of the model in forecasting coarse aerosol concentration. However most desert dust is produced outside the model's domain. Given appropriate boundary conditions, CAMx should transport the dust in its domain and produce good prediction of dust concentration. Do the author's imply that the MACC models provide bad boundary conditions or does CAMx do a poor job transporting the dust within its domain?

3) Section 2.5 should define the uncertainties of the LIRIC algorithm. Several references to evaluation studies are given in the last paragraph, but the authors should briefly present the outcome of these studies, at least to the extent that are relevant for the discussion of their results.

4) The author's definition of PBL is not consistent with the description of the LIRIC algorithm. The authors claim that they search for PBL's top height between "400m and 2.5km". However, LIRIC's lower boundary is set to 600m. In addition LIRIC is "demanding a certain degree of vertical smoothness in the final product", possibly masking the true PBL top. The authors should address these discrepancies and provide estimated of the resulting uncertainties. They should also compare the PBL values derived from LIRIC with the PBL values assumed in the corresponding model profiles.

Technical corrections

1: missing parenthesis " with extensions (CAMx)." This applies also to page 2, line 3.

2: "updated version of the former". This is awkward wording.

13: "For example Mona et al. (2014) compare [..] the dust extinction". Delete "between".

31: "(ENVIRON, 2010)"

Page 3:

27: "Schneider et al. (2000)". The citation seems misplaced and poorly formated.

Page 7:

20: "In the current dataset the full overlap height was calculated at 900m. The lower boundary is set to 600m where the overlap function is still above 90%.". Provide more information about these calculations.

Page 10:

3: How are Q factors calculated?

Many citations are badly formatted and need to be corrected.

Table 2, caption: "The a and c symbols". "a" should be "z".

Fig. 2: What is the meaning of black dots in the HYSPLIT plots?

---

## Referee Comment (RC3) · Anonymous Referee #3 · 13 Feb 2017

The study presents an evaluation of CAMx model against LIRIC output profiles retrieved above the city of Thessaloniki. It is an interesting study with valuable results for the scientific community. However, the authors need to address some issues before publication. As it is currently presented, the idea of the validation is sometimes lost along the manuscript and the paper becomes a little too descriptive. The manuscript would benefit from a more in-depth discussion regarding the validation and more discussion including uncertainties is definitely needed. A review of the writing, which is sometimes confusing, and a possible shortening in length would also be useful to improve the manuscript. Find some more detailed comments below:

[Figure]

Line 7: A fractional bias of 24.8% does not seem "close". I suggest you use the absolute value here instead of percentage.

Line 3: Rephrase this sentence. As it is written, it looks like EMEP is a model instead of a programme.

Lines 27-35: The identification of PM2.5 and PM10 particles with the fine mode and the coarse mode from LIRIC is not completely accurate. Please, rewrite.

Lines 1-9: This information seems more appropriate for the methodology section than for the introduction.

Line 25: "pre-processing"

Line 27: Parenthesis are missing for the reference Schneider et al. (2000). Please, also add the more recent reference Pappalardo et al. (2014)

Line 31: Was the sun photometer deployed at Thessaloniki just for this study?

Line 21: Please, rewrite. It is not clear what you mean by "user defined uncertainties". Does the study by Filioglou et al. (2016) take into the account the uncertainties in the input lidar and radiometer data or just the user defined input parameters? In that case, what is the estimated uncertainty of the output profiles? Include also here that LIRIC has been validated against in-situ aircraft measurements to emphasize that it can be used as an independent reliable tool for the validation of CAMx (see e.g. Granados-Munoz et al., 2016 and Kokkalis et al., 2017)

line 29: What do you mean by characterization procedure of the lidar profiles?

Line 20: How did you calculate the full overlap height? Add references here and/or provide more details.

Line 26: Be more specific for the maximum height, what it is consider a significant quantity?

Line 27: Replace summing by adding

Line 5: Why are you using 1.5 and 2.6 g*cm-3 ? Why don't you use the known aerosol densities provided by CAMx for each case? That would lead to a more accurate comparison between LIRIC and CAMx.

Lines 8-14: Since CAMx lacks of biomass burning aerosol emissions and does not consider desert dust emissions directly, I understand that the fires and dust categories are only used to evaluate the impact that this cases have on the model performance. However, for the evaluation purpose it would make more sense to me to include a category excluding biomass burning and dust cases. That way you would be comparing apples to apples.

Line 21: Please, specify the criteria you use to detect dust cases. Some trajectories do not seem to originate in dust source regions in Figure 1. Idem for continental.

Line 33: This sentence is confusing. Rewrite. What is the diameter for separation between fine and mode in CAMx? Is it the same as in LIRIC?

Line 1: Specify here the number of cases for the comparison. Why does this number emphasize the need of statistics?

Lines 11-12: Provide more updated references.

Line 15: Is this identification criteria based on a sensitivity analysis, previous studies,

etc? Please, explain.

Line 17: What it is the advantage of applying the WCT to LIRIC output profiles instead of the range-corrected signal as in previous studies? Do you obtain similar results using the volume concentration profiles and the RCS?

Line 23: No aerosol is expected above the upper limit in LIRIC, why don't you set these values to zero instead of a constant value?

Line 17: More discussion, including numerical values, is missing here. For the case on January 13, 2014, it looks like most of the aerosol concentration is below the full overlap height. How does this affect the output profiles? How reliable are LIRIC output profiles in this case? Please, add some discussion in this respect.

Lines 19-26: As it is presented, it is not very clear what the contribution of the analysis of the optical properties to the evaluation is. Considering that the goal of the paper is the evaluation of CAMx, I think this section should be shorter or rewritten to clarify its purpose. Additionally, previous studies have shown that backscatter provided by LIRIC is affected by large uncertainties, especially for non-spherical particles (see Wagner et al., 2013 or Granados-Munoz et al., 2014). How do these backscatter profiles compared to those retrieved with a different method ( e.g. Klett-Fernald)?

Line 20: specify if it's extinction or backscatter related Angstrom exponent.

Line 20: It should be figure 2e instead of figure 2d. Include also the CAMx profile in Figure 2f

Lines 10-15: Add numerical values in the discussion. In general in this section 4.1, add more discussion taking into account the uncertainties and shortcomings in LIRIC (and the model if provided by the modellers).

Line 24: Can you provide some information about the boundary layer height values obtained in the study? Besides, because of the incomplete overlap, LIRIC uncertainty in the PBL should be higher than in the troposphere. Take it into account when discussing the results.

Line 7: "are presented"

Line 2: Provide more details on the results obtained removing the dust cases

Line 7: Do you have information about the relative humidity above Thessaloniki during the study period? This could give an idea about how important the hygroscopic growth is and how much it could affect the comparison.

Consider rewriting the conclusions section after all previous comments.

Table 2: Sould be a instead of z (or vice-versa)?

Figure 3: Add also the number of cases for the no fires category in the figure

---

## Author Comment (AC1) · 12 Apr 2017

Response to Referee 2

We would like to thank the reviewer for his/her fruitful comments that helped to improve the manuscript.

General comments: An evaluation study is presented to assess the capability of the air quality model CAMx to describe the aerosol conditions over Thessaloniki. The model simulations are compared to combined sun photometer and lidar observations. Backward trajectories and results of a sophisticated dust forecast model are used to attribute shortcomings to a poor representation of biomass burning and desert dust aerosol. In principle, I like the idea of using different tools, not only measurements, to evaluate the simulations of a specific model and track down shortcomings to suggest model improvements. The evaluation is properly done, although the focus on the comparison with LIRIC data from Thessaloniki only may be too one-sided.

Maybe other observations could be additionally included to underpin the findings.

Unfortunately there weren't any other LIRIC estimates available from lidar stations that are included in the modelling domain.

However, my essential criticism is that the CAMx model is evaluated regarding two aerosol types, which, by design, are not directly computed or only poorly represented. Biomass burning emissions are highly variable in time and space. The actual pollution will largely depend on specific events. Of course, it is not to be expected that the TNO emission database from 2007 in detail is representative for the fire emissions in 2013 – 2015. The same holds for Saharan dust that is not online computed based on modelled winds but input as boundary condition. This must be considered when evaluating the model results, and the conclusions have to be revised in this regard. How exactly is the CAMx model suggested to be improved with this in mind, and based on the evaluation results?

Our aim was not to evaluate CAMx for its performance regarding smoke and desert dust. At a first step we tried to use all available measurements for the period under study in order to investigate whether there is a good agreement between the model and the LIRIC estimates. From the analysis we concluded that the agreement is very good for fine mode aerosols excluding the smoke incidents and dust events, which means that most other sources (anthropogenic and natural) are reasonably represented in the model. Concerning the smoke we suggest that we cannot expect an agreement, since the emission inventory has not such on-line module. Concerning the

Saharan dust, indeed any desert dust in CAMx simulations results from the boundary conditions. Following the reviewer's suggestion we examined maps of CAMx for selective cases that were affected by desert dust and it seems that, for some of them there are issues in the transportation of CAMx PM2.5-10 from the boundaries to long distances. However, the small number of cases available for such an analysis does not allow to draw firm conclusions on this issue, especially to distinguish what is the main issue, the boundaries themselves or the transport. A relevant discussion is added in sections 4.1 and 4.2 and in the conclusions. The figures 3c and 3d and 6 on the left as well as tables 5 and 6 were also modified.

Specific comments:

1. Page 4, line 10: A plot showing the model domains would be very helpful, in particular, to show if relevant Saharan dust sources are included.

The domains of CAMx have been included in the manuscript (Figure 1). The figure numbering has been adjusted in the text.

The text has been modified according to the reviewer's suggestion: "The domains of CAMx are presented in figure 1."

2. Page 11, lines 5 – 8: Here and later in Section 4, the study period 2013 – 2015 should be mentioned in order to clearly separate example cases from the broader statistical analysis.

The text was modified according to the reviewers suggestion:

Page 11, line 5: "An ensemble of 24 measurements in the period 2013-2015."

Page 11, line 10: "In this section the simulated profiles of CAMx are compared against the observational profiles of LIRIC in the period 2013-2015."

3. Figures 1 and 3 – 7: Please indicate in each figure caption whether the results refer to a specific case or the entire period 2013 – 2015.

The text was modified according to the reviewers suggestion.

The phrase "the period 2013-2015" was added to all the aforementioned figures.

---

## Author Comment (AC2) · 12 Apr 2017

Response to Referee 1

We would like to thank the reviewer for his/her fruitful comments that helped to improve the manuscript.

The paper of Siomos et al. presents an interesting example of aerosol model evaluation based on remote sensing observations. The manuscripts highlights the potential and pitfalls for such a comparison, therefore it could be of interest for the wider atmospheric community. The manuscript is worth publishing after addressing several comments listed below.

General comments

1) A main issue with the presented analysis is that the authors compare fine and coarse particles defined in two fundamentally different ways. According to the text, the model's fine mode is defined as particles with aerodynamic diameter less than 2.5um, while LIRIC's fine mode is defined as particles with (optical) diameter less than ~0.4 – 1.2 um. Before this study is published, the authors should thoroughly discuss this issue and justify why their comparison gives any meaningful results.

The reviewer is right. In the current analysis the PM2.5 particles should include all the fine particles and a small part of the coarse particles that is variable depending on the case. After analyzing the size distribution of all the cases, we found that this fraction of the coarse mode ranges from 5-25%. It is possible to convert the fine and coarse modes of LIRIC to PM2.5 and PM2.5-10 particles using this fraction. In the LIRIC inversion, the normalized size distribution of each mode is derived from the columnar size distribution for each height bin, resulting in constant extinction and backscattering efficiencies per aerosol mode. Taking that into account, the fraction of the sunphotometer's coarse mode that belongs in the PM2.5 region is independent of the height. Thus, it is possible to use this fraction in order to convert the LIRIC fine and coarse profiles to PM2.5 and PM2.5-10 profiles that are consistent with CAMx, by subtracting (for each individual case) the PM2.5 coarse fraction from each LIRIC coarse profile and adding it to the respective LIRIC fine profile. This affects the LIRIC fine and coarse concentration and integrated mass values as well as the fine center of mass values. The manuscript has been updated accordingly. The "fine" and "coarse" terminology has been replaced by "PM2.5" and "PM2.5-10" where it was necessary. The tables 3, 4, 6 and figures 3, 4, 5, 6, 7 are modified. The discussion and is also

modified accordingly. The following paragraphs have been added to the text to describe this methodology.

The following text was added at the end of Section 3.1.1: "Another hindrance in the analysis is that the fine and coarse mode of LIRIC are not directly comparable with the PM2.5 and PM2.5-10 modes of CAMx. The PM2.5 particles should include all the fine particles and a small part of the coarse particles that changes depending on the case. Additionally, the size distribution of the sunphotometer usually surpasses the PM10 diameter limit. Fortunately, it is possible to convert the fine and coarse modes of LIRIC to PM2.5 and PM2.5-10 particles. In the LIRIC inversion, the normalized volume size distribution of each mode is derived by separating the columnar size distribution of the sunphotometer in the two modes. The normalized distribution of each mode remains constant with height. Taking that into account, the fractions of the sunphotometer's coarse mode that belong in the PM2.5 region and the region outside the PM10 particles can be calculated from the sunphotometer's volume size distribution. Then, the fine and coarse concentration profiles of each LIRIC case can be converted to PM2.5 and PM2.5-10 profiles using the equations 2 and 3. "

The new equations are presented below:

$$c_{PM2.5}(z) = c_{fine}(z) + c_{coarse}(z) \cdot \frac{\int_{r_{f-c}}^{r_{PM2.5}} \frac{dV}{dr} \cdot dr}{\int_{r_{f-c}}^{r_c} \frac{dV}{dr} \cdot dr}$$

$$c_{PM2.5-10}(z) = c_{coarse}(z) - c_{coarse}(z) \cdot \frac{\int_{r_{f-c}}^{r_{PM2.5}} \frac{dV}{dr} \cdot dr + \int_{r_{PM10}}^{r_c} \frac{dV}{dr} \cdot dr}{\int_{r_{f-c}}^{r_c} \frac{dV}{dr} \cdot dr})$$

Where $c_{fine}$ , $c_{coarse}$ , $c_{PM2.5}$ , $c_{PM2.5-10}$ are the concentration profiles of LIRIC before and after the conversion and dV/dr is the aerosol volume size distribution of the sunphotometer as a function of the aerosol radius. The radii $r_c$ , $r_{f-c}$ , $r_{PM2.5}$ , $r_{PM10}$ are in μm units and correspond to the upper limit of the sunphotometer size distribution, the separator radius between the fine and the coarse mode of the sunphotometer and the PM2.5 and the PM10 separator radii respectively.

Concerning the optical versus the aerodynamic diameter, it is possible to convert from one type to the other (C.-H. Chien et al. / Journal of Aerosol Science 101 (2016) 77–85). Using their formula for the NaCl particles, an aerodynamic diameter of 2.5um

corresponds to an optical diameter of approximately 2.0um. However, we aren't going to perform this conversion for the following reasons.
The aerosol concentration in CAMx depends on the emissions within the domain and the boundary conditions. In both cases, the species concentration is imported from external models (TNO emissions, ECMWF emissions, NEMO). In general, the aerosol concentration in models is based on satellite and ground based measurements. Taking this into account, it is difficult to characterize the aerosol diameter of a model as exclusively optical or aerodynamic. Even the particles that are produced from chemical reactions inside CAMx (i.e. the secondary organic compounds) do not carry the information of a detailed size distribution. They are just flagged as PM2.5 or PM2.5-10. As a result, we removed the word "aerodynamic" from the manuscript since it is misleading.

2) Desert dust is included in the model only as a boundary conditions and this explains, according the the authors, the poor performance of the model in forecasting coarse aerosol concentration. However most desert dust is produced outside the model's domain. Given appropriate boundary conditions, CAMx should transport the dust in its domain and produce good prediction of dust concentration. Do the author's imply that the MACC models provide bad boundary conditions or does CAMx do a poor job transporting the dust within its domain?

Following the reviewer's suggestion we examined maps of CAMx for selective cases that were affected by desert dust and it seems that, for some of them there are issues in the transportation of CAMx PM2.5-10 from the boundaries to long distances. Taking into account the number of dust cases in this study, it is difficult to draw a firm conclusion on the prevalent source of bias. This can be examined in a future study. Consequently, the text in section 4.1, 4.2 and in the conclusions is modified so that both the lack of dust emissions in the domain (other than the boundary conditions), and the model's transportation of dust are presented as potential sources of bias in the dust concentration.

During the analysis we detected a bug in our algorithms where the "soil PM2.5" and the "soil PM2.5-10" components were in some cases identified as the "other PM2.5" and the "other PM2.5-10" components respectively and vice versa. This is now corrected and the relevant discussion in both sections 4.1 and 4.2 is modified accordingly. The new figures 4c, 4d and 7 (left) as well as tables 5 and 6 are modified accordingly.

In addition, after reinspecting the data processing algorithms, we noticed that for the center of mass calculation the vertical resolution of the profiles was considered

constant which is not the case when the model's eta levels are considered. We recalculated the center of mass with variable vertical resolution and tables 3 and 5 and figures 4 and 7a have been modified accordingly.

3) Section 2.5 should define the uncertainties of the LIRIC algorithm. Several references to evaluation studies are given in the last paragraph, but the authors should briefly present the outcome of these studies, at least to the extent that are relevant for the discussion of their results.

The text was modified to: "The effects of multiple user defined uncertainties, such as the upper and lower limit heights of the profile and the algorithm's regularization parameters, on the final result has been studied by Granados-Muñoz et al. (2014) and Filioglou et al. (2017) for selective case studies in Granada and Thessaloniki respectively. They agree that the parameter that produces the biggest uncertainties is the lower limit height of the profile. Furthermore, the LIRIC retrievals have already been evaluated for volcanic and desert dust particles by Wagner et al. (2013) showing that the inversion can be accurate for two quite different types of aerosol. The aerosol extinction products of LIRIC has also been compared against the respective products from the Generalized Aerosol Retrieval from Radiometer and Lidar Combined data (GARRLiC) algorithm and against the retrievals from raman lidar measurements (Bovchaliuk et al., 2016). Finally, LIRIC has also been validated against in-situ aircraft measurements (e.g., Granados-Muñoz et al., 2016a; Kokkalis et al., 2017). Granados-Muñoz et al. (2016a) compared the LIRIC retrievals with airborn in-situ measurements and found a promising agreement with the differences between the two staying within the expected uncertainties. Kokkalis et al. (2017) analyzed a biomass burning case. Their comparison between the LIRIC retrievals and the aircraft measurements resulted in a good performance of the algorithm for the fine particles. As a result it can be used as an independent reliable tool for the validation of CAMx."

4) The author's definition of PBL is not consistent with the description of the LIRIC algorithm. The authors claim that they search for PBL's top height between "400m and 2.5km". However, LIRIC's lower boundary is set to 600m.

Indeed this is not clear in the text. The overlap correction is applied normally but it is still necessary to limit the profiles since the overlap function can't be trusted down to the ground. For the comparison between LIRIC and CAMx we chose to limit the profiles in a region where the overlap function is above 0.9 (600m) instead of 0.7 that we typically use in the lidar data processing in order to reduce the uncertainty of the

overlap correction. The 600m limit also apply to the PBL height retrieval. The 400m is a typo and it will be corrected.

The text was modified according to the reviewer's suggestions: "Identification criteria are necessary for the selection of the PBL height. The top of the layer between 600m and 2.5km with the minimum value in the transformed signal is chosen as the boundary layer height."

In addition LIRIC is "demanding a certain degree of vertical smoothness in the final product", possibly masking the true PBL top. The authors should address these discrepancies and provide estimated of the resulting uncertainties.

By comparing the Klett lidar backscatter profiles from our operational algorithms and the LIRIC backscatter profiles using the concentration and the backscattering efficiencies from LIRIC (see equation) we have seen that the vertical structure is similar, especially for strong layers such as the boundary layer.

They should also compare the PBL values derived from LIRIC with the PBL values assumed in the corresponding model profiles.

Since the vertical resolution is quite lower in CAMx than in LIRIC (eta levels against a constant vertical resolution of 15m) it would be pointless to apply the WCT or similar techniques that take advantage of the aerosol vertical distribution to the profiles of CAMx.

Technical corrections

Line 1: missing parenthesis " with extensions (CAMx)." This applies also to page 2, line 3.

The text was changed according to the reviewer's suggestion.

Line 2: "updated version of the former". This is awkward wording.

The text was rephrased to: "the Dust Regional Atmospheric Model (BSC-DREAM8b) were deployed"

Line 13: "For example Mona et al. (2014) compare [..] the dust extinction". Delete "between".

The text was changed according to the reviewer's suggestion.

Line 31: "(ENVIRON, 2010)"

The text was changed according to the reviewer's suggestion.

Line 27: "Schneider et al. (2000)". The citation seems misplaced and poorly formated.

The text was changed to: "(EARLINET) (Schneider et al., 2000; Papalardo et al., 2014)"

Line 20: "In the current dataset the full overlap height was calculated at 900m. The lower boundary is set to 600m where the overlap function is still above 90%.". Provide more information about these calculations.

See the relative comment response in Reviewer 3.

Page 7, Line 20: The text has been modified to: "A lower height boundary has to be determined due to the overlap function of the lidar system. Operationally, we apply the method of Wandinger et al. 2002 for the calculation of the overlap function and the full overlap height. In the current dataset the full overlap height was calculated at 900m. The correction however cannot be trusted down to the ground (Wandinger et al. 2002). In this study, we apply the correction down to 600m where the overlap function is still above 90% and use this height as the lower boundary of the LIRIC inversion. Below this height the lidar signals are considered constant during the LIRIC inversion. The concentration retrievals are also kept constant below 600m."

Line 3: How are Q factors calculated?

The text was rephrased to: "where Q ext is the extinction efficiency and Q bsc is the backscattering efficiency calculated by LIRIC."

Many citations are badly formatted and need to be corrected.

The urls in the citations were removed. Some empty fields were also cleared.

Table 2, caption: "The a and c symbols". "a" should be "z".

The text was changed according to the reviewer's suggestion.

Fig. 2: What is the meaning of black dots in the HYSPLIT plots?

The following sentence was added in the caption of Figure 2: "The big black dots in a and b indicate 24h intervals"

---

## Author Comment (AC3) · 12 Apr 2017

Response to Referee 3

We would like to thank the reviewer for his/her fruitful comments that helped to improve the manuscript.

The study presents an evaluation of CAMx model against LIRIC output profiles retrieved above the city of Thessaloniki. It is an interesting study with valuable results for the scientific community. However, the authors need to address some issues before publication. As it is currently presented, the idea of the validation is sometimes lost along the manuscript and the paper becomes a little too descriptive. The manuscript would benefit from a more in-depth discussion regarding the validation and more discussion including uncertainties is definitely needed. A review of the writing, which is sometimes confusing, and a possible shortening in length would also be useful to improve the manuscript. Find some more detailed comments below:

Line 7: A fractional bias of 24.8% does not seem "close". I suggest you use the absolute value here instead of percentage.

The text has been modified to: "mean bias of 0.57 km."

Line 3: Rephrase this sentence. As it is written, it looks like EMEP is a model instead of a programme.

The following text was removed: "European Monitoring and Evaluation Programme EMEP"

Lines 27-35: The identification of PM2.5 and PM10 particles with the fine mode and the coarse mode from LIRIC is not completely accurate. Please, rewrite.

See the relative comment response in Reviewer 1.

Page 2, Line 34: The text was modified to: "Instead of evaluating the performance of CAMx only for the PM10 particles, we separate the fine from the coarse particles by applying the LIRIC technique, then we convert the fine and coarse concentration profiles of LIRIC to PM2.5 and PM2.5-10 profiles and perform the validation for the PM2.5 and PM2.5-10 individually."

Lines 1-9: This information seems more appropriate for the methodology section than for the introduction.

In the introduction we briefly mention the tools used in our study which are described in more detail in the methodology section.

Line 25: "pre-processing"

The text has been modified according to the reviewer's suggestion.

Line 27: Parenthesis are missing for the reference Schneider et al. (2000). Please, also add the more recent reference Pappalardo et al. (2014)

The text was changed to: "(EARLINET) (Schneider et al., 2000; Papalardo et al., 2014)"

Line 31: Was the sun photometer deployed at Thessaloniki just for this study?

The lines 31-32 have been rephrased to: "We used measurements from a CIMEL multiband sun-sky photometer which  was installed in Thessaloniki in 2003 as part of the AERONET Global Network."

Line 21: Please, rewrite. It is not clear what you mean by "user defined uncertainties". Does the study by Filioglou et al. (2016) take into the account the uncertainties in the input lidar and radiometer data or just the user defined input parameters? In that case, what is the estimated uncertainty of the output profiles? Include also here that LIRIC has been validated against in-situ aircraft measurements to emphasize that it can be used as an independent reliable tool for the validation of CAMx (see e.g. Granados-Munoz et al., 2016 and Kokkalis et al., 2017)

See the relative comment response to Reviewer 1.

The text was modified to: "The effects of multiple user defined uncertainties, such as the upper and lower limit heights of the profile and the algorithm's regularization parameters, on the final result has been studied by Granados-Muñoz et al. (2014) and Filioglou et al. (2017) for selective case studies in Granada and Thessaloniki respectively. They agree that the parameter that produces the biggest uncertainties is

the lower limit height of the profile. Furthermore, the LIRIC retrievals have already been evaluated for volcanic and desert dust particles by Wagner et al. (2013) showing that the inversion can be accurate for two quite different types of aerosol. The aerosol extinction products of LIRIC has also been compared against the respective products from the Generalized Aerosol Retrieval from Radiometer and Lidar Combined data (GARRLiC) algorithm and against the retrievals from raman lidar measurements (Bovchaliuk et al., 2016). Finally, LIRIC has also been validated against in-situ aircraft measurements (e.g., Granados-Muñoz et al., 2016a; Kokkalis et al., 2017). Granados-Muñoz et al. (2016a) compared the LIRIC retrievals with airborn in-situ measurements and found a promising agreement with the differences between the two staying within the expected uncertainties. Kokkalis et al. (2017) analyzed a biomass burning case. Their comparison between the LIRIC retrievals and the aircraft measurements resulted in a good performance of the algorithm for the fine particles. As a result it can be used as an independent reliable tool for the validation of CAMx."

line 29: What do you mean by characterization procedure of the lidar profiles?

The text has been changed to: "aerosol type identification"

Line 20: How did you calculate the full overlap height? Add references here and/or provide more details.

From the method of Wandinger et al. 2002 both the overlap function and the full overlap height are calculated. In this study we have applied the overlap correction per case using a typical overlap function. The overlap correction, however, cannot be extended to the ground. Typically, we limit the profile at the height where the function is higher than 0.7. For the CAMx validation however we preferred to use 0.9 (600m) to be on the safe side. This not clear in the text and it will be added.

Furthermore, in the original analysis we kept the lidar signals constant below 600m during the LIRIC inversion but the concentration product of LIRIC can be slightly variable even below the lower limit. To be entirely consistent with the idea of constant products below 600m we decided to keep the concentration profiles constant below this lower limit. This slightly affects the figures 3,4,5,6 and the tables 3,4,5,6. The text was also modified in order to clarify this adjustment.

Page 7, Line 20: The text has been modified to: "A lower height boundary has to be determined due to the overlap function of the lidar system. Operationally, we apply the

method of Wandinger et al. 2002 for the calculation of the overlap function and the full overlap height. In the current dataset the full overlap height was calculated at 900m. The correction however cannot be trusted down to the ground (Wandinger et al. 2002). In this study, we apply the correction down to 600m where the overlap function is still above 90% and use this height as the lower boundary of the LIRIC inversion. Below this height the lidar signals are considered constant during the LIRIC inversion. The concentration retrievals are also kept constant below 600m."

Line 26: Be more specific for the maximum height, what it is consider a significant quantity?

The text was rephrased to: "The upper boundary depends on the maximum height where aerosol exist in a significant quantity, that is, a region where the lidar signal from the aerosol backscattering can no longer be separated from the noise. This height can vary depending on the atmospheric conditions."

Line 27: Replace summing by adding

The text was modified according to the reviewer's suggestion.

Line 5: Why are you using 1.5 and 2.6 g*cm-3? Why don't you use the known aerosol densities provided by CAMx for each case? That would lead to a more accurate comparison between LIRIC and CAMx.

The use of a CAMx derived density on the LIRIC profiles presupposes that the mixing ratio of each species is well predicted by CAMx. Otherwise the reference data of LIRIC would be affected by uncertainties originating from CAMx. Thus we preferred to use constant conversion values that are commonly used in the literature. This is also the way of Binietoglou et al. 2015. A more direct comparison would be to convert the CAMx profiles to ppbv. This conversion, however, was rejected after testing it because we wanted to avoid confusion by using a unit that is not adopted by the modeler's community like ppbv and the results were also pretty similar.

Lines 8-14: Since CAMx lacks of biomass burning aerosol emissions and does not consider desert dust emissions directly, I understand that the fires and dust categories are only used to evaluate the impact that this cases have on the model performance. However, for the evaluation purpose it would make more sense to me to include a

category excluding biomass burning and dust cases. That way you would be comparing apples to apples.

The main reason that we didn't isolate the cases that aren't biomass burning and dust is that our dataset is limited. By removing the 6 dust cases from the "non fires" group, the dataset is reduced to 11 measurements. Furthermore, by checking the dust cases individually we observed that unlike the coarse mode, the fine mode is generally in good agreement between LIRIC and CAMx. In order to provide more information on the dust cases the old Figure 4 is modified and the dust cases are displayed with orange color.

Line 21: Please, specify the criteria you use to detect dust cases. Some trajectories do not seem to originate in dust source regions in Figure 1. Idem for continental.

To characterize the cases we check the trajectories separately in the PBL and the FT. Then, one trajectory, either in the PBL or in the FT, is required in order to identify the measurement as dust or biomass burning. Additionally, an empirical criterion of a maximum dust concentration above 10ugr/m^3 in the DREAM profile is also applied to ensure that the trajectory carries dust.

Line 33: This sentence is confusing. Rewrite. What is the diameter for separation between fine and mode in CAMx? Is it the same as in LIRIC?

See the relative comment response to Reviewer 1.

The text has been modified according to the reviewer's suggestions: "Another hindrance in the analysis is that the fine and coarse mode of LIRIC are not directly comparable with the PM2.5 and PM2.5-10 modes of CAMx. The PM2.5 particles should include all the fine particles and a small part of the coarse particles that changes depending on the case. Additionally, the size distribution of the sunphotometer usually surpasses the PM10 diameter limit. Fortunately, it is possible to convert the fine and coarse modes of LIRIC to PM2.5 and PM2.5-10 particles. In the LIRIC inversion, the normalized volume size distribution of each mode is derived by separating the columnar size distribution of the sunphotometer in the two modes. The normalized distribution of each mode remains constant with height. Taking that into account, the fractions of the sunphotometer's coarse mode that belong in the PM2.5 region and the region outside the PM10 particles can be calculated from the sunphotometer's volume

size distribution. Then, the fine and coarse concentration profiles of each LIRIC case can be converted to PM2.5 and PM2.5-10 profiles using the equations 2 and 3. "

Line 1: Specify here the number of cases for the comparison. Why does this number emphasize the need of statistics?

The text was modified according to the reviewer's suggestions: "A total of 22 cases take part in the comparison. We preferred a statistical approach in the analysis rather than comparing each case individually since the size of the dataset permits it."

Lines 11-12: Provide more updated references.

The text was modified to: "(e.g., Flamant et al., 1997; Menut et al., 1999; Brooks, 2003; Tomasi and Perrone, 2006; Bravo-Aranda et al., 2016)

Line 15: Is this identification criteria based on a sensitivity analysis, previous studies, etc? Please, explain.

The reason why we use the upper limit criteria is that by applying the WTC method it is possible that a strong elevated layer could be identified as the PBL. This is also specified in Baars et al., 2008. In one of the cases they analyzed, an elevated dust layer complicated the derivation of the PBL top. Garrett, 1992  mention that the ABL typically extends from the ground to 2–3 km.  Additionally, Georgoulias et al. 2009 in their study show that for noon measurements the mixing layer top is most of the time below 2600m for Thessaloniki. The selection of the upper limit value at 2500m is based on these studies.

The lower boundary corresponds to the height where the overlap function of the lidar system is above 0.9, that is 600m. The value of 400m is a typo and will be corrected. This is also mentioned in the response to Reviewer 1.

The text is modified according to the reviewer's suggestion "Identification criteria are necessary for the selection of the PBL height. The top of the layer between 600m and 2.5km with the minimum value in the transformed signal is chosen as the boundary layer height. The upper limit is necessary in order to avoid identifying the top of sharp elevated layers as the PBL. According to Georgoulias et al. (2009) the upper limit of 2.5km is realistic for Thessaloniki. Baars et al. (2008) presented a case where an

elevated dust layer complicated the PBL height retrieval with the WCT method. The wavelet transform is applied to the LIRIC concentration profiles before the upscaling of the resolution."

Line 17: What it is the advantage of applying the WCT to LIRIC output profiles instead of the range-corrected signal as in previous studies?

The range-corrected signal is an optical product that is typically used for the boundary layer height calculation because it is representative of the aerosol quantity and it is also much more straightforward to calculate than i.e. the aerosol backscatter or the aerosol extinction coefficient. Here, the aerosol concentration is already available and it provides direct information of the quantity of the aerosols. Thus, we preferred to use the LIRIC products instead.

Do you obtain similar results using the volume concentration profiles and the RCS?

The application of the WCT either in the range-corrected signal or in the LIRIC concentration provides similar results.

Line 23: No aerosol is expected above the upper limit in LIRIC, why don't you set these values to zero instead of a constant value?

The vertical profiles of CAMx extend up to 9.5km. As it can be seen from figures 3a and 3b the model typically provides non zero values in the whole profile so it will not be realistic to assume that the concentration is zero above the upper limit. Consequently, we use the information of the last point of the profile as the best guess of the aerosol load above that height. In figures 3a and 3b it can be seen that this choice produced a mean concentration of 1-2ugr/m^-3 at 10km for both the fine and the coarse particles which is not abnormal for this atmospheric region.
However, it is true that in rare occasions when the SNR in the lidar signals is quite low, especially near the upper limit, the LIRIC inversion can be affected. Higher than expected concentration values can be produced near the upper limit resulting in unrealistic LIRIC overestimations for the integrated mass values in the FT. We detected that this is the case for two measurements in the current dataset, one that belongs to the "continental" category and one that belongs in the "fires" category. In order to ensure the quality of our reference data we decided to remove those two cases from

the analysis. This reduces the total number of cases to 22, the number of the "fires" cases to 5 and the number of the "non fires" cases to 16.

Line 17: More discussion, including numerical values, is missing here. For the case on January 13, 2014, it looks like most of the aerosol concentration is below the full overlap height. How does this affect the output profiles? How reliable are LIRIC output profiles in this case? Please, add some discussion in this respect.

See the relative comment response in Reviewer 1

Lines 19-26: As it is presented, it is not very clear what the contribution of the analysis of the optical properties to the evaluation is. Considering that the goal of the paper is the evaluation of CAMx, I think this section should be shorter or rewritten to clarify its purpose. Additionally, previous studies have shown that backscatter provided by LIRIC is affected by large uncertainties, especially for non-spherical particles (see Wagner et al., 2013 or Granados-Munoz et al., 2014). How do these backscatter profiles compared to those retrieved with a different method ( e.g. Klett-Fernald)?

The agreement between LIRIC and Klett derived optical properties is very good. We present in the paper two typical cases as a demonstration of the methodology used to examine the aerosol profiles for each individual case. The inclusion of lidar ratio and angstrom exponent profiles provides further evidence for identifying different aerosol types and layers, but such profiles can be misleading if someone does not examine in parallel the extinction and backscatter profiles. A relevant comment has been added in the text

Add text

Line 20: specify if it's extinction or backscatter related Angstrom exponent.

It's the extinction Angstrom exponent. The text was modified according to the reviewer's suggestions.

Line 20: It should be figure 2e instead of figure 2d. Include also the CAMx profile in Figure 2f

The figures have been renamed according to the reviewer's suggestion.

The CAMx profile in figure 2f is not provided since it is biased in a similar way with figure 2d. This is specified in the text. The space inside this small figure is also limited. Furthermore, the main point in this section is to shortly demonstrate the capabilities and all the possible products of LIRIC for two different aerosol cases.

Lines 10-15: Add numerical values in the discussion. In general in this section 4.1, add more discussion taking into account the uncertainties and shortcomings in LIRIC (and the model if provided by the modellers).

Section 4.1 was modified according to the reviewer's suggestions

Line 24: Can you provide some information about the boundary layer height values obtained in the study?

The boundary layer height retrievals of the cases vary between 600m and 2500m without showing any strong pattern. A slight preference for PBL values in the range 1000-1500m can be observed. However, one has to take into account that the cases are not uniformly distributed either in the annual and daily cycle. Both of these variables highly affect the PBL height.

Besides, because of the incomplete overlap, LIRIC uncertainty in the PBL should be higher than in the troposphere. Take it into account when discussing the results.

As it was mentioned in the previous comment responses we will include in the text that the lidar signals are overlap corrected down to 600m since it was not clearly specified. Consequently, the signals can be trusted down to 600m. Indeed the missing part of the signal (0-600m) that is assumed to be constant can produce uncertainties in the retrieval. Munoz et al. 2014 have studied the uncertainty of the LIRIC retrieval using different parts of the signal that were not overlap corrected, and thus always underestimated, within acceptable overlap values (above 0.8). They found that the produced uncertainty is higher in the near range in terms of absolute values. This approach, however, includes both the uncertainty of the part of signals that is not overlap corrected and the uncertainty of the assumption of constant signals below the lower limit. For that reason, it is uncertain if the height variability that they observe applies to our case.

Line 7: "are presented"

The text was modified according to the reviewer's suggestion.

Line 2: Provide more details on the results obtained removing the dust cases

The following text was added: "The comparison for the PM2.5 particles is actually affected in a negative way due to the limited number of measurements in the dataset."

The dust cases in all the scatterplots will also be marked with an orange color.

Line 7: Do you have information about the relative humidity above Thessaloniki during the study period? This could give an idea about how important the hygroscopic growth is and how much it could affect the comparison. Consider rewriting the conclusions section after all previous comments.

Unfortunately, the only information available for this period is the water content which is added in the PM2.5 calculation. This could be analyzed in a future study.

Table 2: Should be a instead of z (or vice-versa)?

See the relative comment response to Reviewer 1

Figure 3: Add also the number of cases for the no fires category in the figure

Figure 3 has been updated according to the reviewer's suggestions.